# An Online Learning Theory of Trading-Volume Maximization

**Tommaso Cesari**
EECS
University of Ottawa
Ottawa, Canada
`tcesari@uottawa.ca`

**Roberto Colomboni**
DEIB / Dept. of Computer Science
Politecnico di Milano / University of Milan
Milano, Italy
`robertocolomboni@polimi.it`

## Abstract

We explore brokerage between traders in an online learning framework. At any round $t$, two traders meet to exchange an asset, provided the exchange is mutually beneficial. The broker proposes a trading price, and each trader tries to sell their asset or buy the asset from the other party, depending on whether the price is higher or lower than their private valuations. A trade happens if one trader is willing to sell and the other is willing to buy at the proposed price. Previous work provided guidance to a broker aiming at enhancing traders' total earnings by maximizing the *gain from trade*, defined as the sum of the traders' net utilities after each interaction. This classical notion of reward can be highly unfair to traders with small profit margins, and far from the real-life utility of the broker. For these reasons, we investigate how the broker should behave to maximize the trading volume, i.e., the *total number of trades*. We model the traders' valuations as an i.i.d. process with an unknown distribution. If the traders' valuations are revealed after each interaction (full-feedback), and the traders' valuations cumulative distribution function (cdf) is continuous, we provide an algorithm achieving logarithmic regret and show its optimality up to constants. If only their willingness to sell or buy at the proposed price is revealed after each interaction (2-bit feedback), we provide an algorithm achieving poly-logarithmic regret when the traders' valuations cdf is Lipschitz and show its near-optimality. We complement our results by analyzing the implications of dropping the regularity assumptions on the unknown traders' valuations cdf. If we drop the continuous cdf assumption, the regret rate degrades to $\Theta(\sqrt{T})$ in the full-feedback case, where $T$ is the time horizon. If we drop the Lipschitz cdf assumption, learning becomes impossible in the 2-bit feedback case.

## 1 Introduction

In modern financial markets, Over-the-Counter (OTC) trading platforms have emerged as dynamic and decentralized hubs, offering diverse alternatives to traditional exchanges. In recent years, these markets have experienced remarkable growth, solidifying their central role in the global financial ecosystem: OTC asset trading in the US surpassed 50 trillion USD in value in 2020 (Weill, 2020), with an upward trend documented since 2016 (www.bis.org, 2022).

Brokers play a crucial role in OTC markets. Beyond acting as intermediaries between traders, they utilize their understanding of the market to identify the optimal prices for assets. Additionally, traders in these markets often respond to price changes: higher prices usually lead to selling, while lower prices typically result in buying (Sherstyuk et al., 2020). This adaptability appears across various asset classes, including stocks, derivatives, art, collectibles, precious metals and minerals, energy commodities (like gas and oil), and digital currencies (cryptocurrencies) (Bolić et al., 2024).

Our study draws inspiration from recent research analyzing the bilateral trade problem from an online learning perspective (Cesa-Bianchi et al., 2021; Azar et al., 2022; Cesa-Bianchi et al., 2023; 2024a; Bolić et al., 2024; Bernasconi et al., 2024; Bachoc et al., 2024a;b). In particular, we build on insights from Bolić et al. (2024), which addresses the brokerage problem in OTC markets where traders may decide to buy or sell their assets depending on prevailing market conditions.

## 1.1 MOTIVATIONS FOR CHOOSING TRADING VOLUME AS REWARD

Previous works have entirely focused on scenarios where brokers aim at maximizing the so-called cumulative *gain from trade*—the sum of the net utilities of the traders over the entire sequence of interactions with the broker. This classical approach has the two following pitfalls.

**Traders' Perspective.**  Gain-from-trade maximization can cause unfairness in settings where the majority of traders make a living off of small margins (e.g., in micro trading or high-frequency trading), and only a handful of high-payoff trades have the potential to occur. In these cases, gain-from-trade maximization can lead to sacrificing the majority of the population in favor of a small minority of traders that are lucky enough to be paired with people that are willing to be greatly underpaid for the good on sale. In contrast, trading-volume maximization gives the same dignity to all traders, granting everybody the same opportunity to trade, independently of their buying power. For a striking concrete example of this pitfall, see Section 3.

**Broker's Perspective.**  From the broker's perspective, too, it might not be as beneficial to potentially miss out on traders' exchanges by maximizing the gain from trade, given that, typically, brokers only earn when trades occur. For example, in settings where traders have to pay a small fee for each trade, it is clear that the broker's ultimate goal is to maximize trading volume. Another example where maximizing trading volume is superior to maximizing the gain from trade is the one discussed in the Trader's Perspective paragraph (and Section 3). In this case, a gain-from-trade maximizing broker would risk alienating the vast majority of the population which, realistically, would end up leaving a broker that does not give them trading opportunities, consequently hurting the broker's bottom line.

For these reasons, in this work, we aim at providing strategies that boost the trading volume by maximizing the *number of trades* in the broker-traders interaction sequence.

## 1.2 SETTING

In what follows, for any two real numbers $a, b$, we denote their minimum by $a \wedge b$ and their maximum by $a \vee b$. We now describe the brokerage online learning protocol.

For any time $t = 1, 2, \dots$

- Two traders arrive with their private valuations $V_{2t-1}$ and $V_{2t}$
- The broker proposes a trading price $P_t$
- If the price $P_t$ is between the lowest valuation $V_{2t-1} \wedge V_{2t}$ and the highest valuation $V_{2t-1} \vee V_{2t}$—meaning the trader with the lower valuation is willing to sell at $P_t$ and the trader with the higher valuation is willing to buy at $P_t$—the transaction occurs with the higher-valuation trader purchasing the asset from the lower-valuation trader at the price $P_t$
- The broker receives some feedback

As commonly assumed in the existing bilateral trade literature, we assume valuations and prices belong to $[0, 1]$. While previous literature aims at maximizing the cumulative *gain from trade*— defined as the sum of traders' net utilities[1] in the whole interaction sequence—our objective is to maximize the *number of trades*. Formally, for any $p, v_1, v_2 \in [0, 1]$, our utility posting a price $p$ when the valuations of the traders are $v_1$ and $v_2$ is

$$g(p, v_1, v_2) \coloneqq \mathbb{I}\{v_1 \wedge v_2 \le p \le v_1 \vee v_2\} \ .$$

The goal of the broker is to minimize the *regret*, defined, for any time horizon $T \in \mathbb{N}$, as

$$R_T \coloneqq \sup_{p \in [0,1]} \mathbb{E}\left[\sum_{t=1}^{T} \big(G_t(p) - G_t(P_t)\big)\right],$$

where $G_t(q) \coloneqq g(q, V_{2t-1}, V_{2t})$ for all $q \in [0, 1]$ and $t \in \mathbb{N}$, and the expectation is taken over the randomness present in $(V_t)_{t \in \mathbb{N}}$ and the (possible) randomness used by the broker's algorithm to generate the prices $(P_t)_{t \in \mathbb{N}}$.

---

[1]Formally, for any $p, v_1, v_2 \in [0, 1]$, the gain from trade of a price $p$ when the valuations of the traders are $v_1$ and $v_2$ is $\text{GFT}(p, v_1, v_2) \coloneqq (v_1 \vee v_2 - v_1 \wedge v_2)\,\mathbb{I}\{v_1 \wedge v_2 \le p \le v_1 \vee v_2\}$.

| | $M$-Lipschitz | Continuous | General |
|---|---|---|---|
| Full | $\Omega(\ln T)$ Thm 2 | $O(\ln T)$ Thm 1 | $\Theta\big(\sqrt{T}\big)$ Thm 5+6 |
| 2-Bit | $O\big(\ln(MT)\ln T\big), \Omega\big(\ln(MT)\big)$ Thm 3+4 | $\Omega(T)$ Thm 7 | $\Omega(T)$ |

Table 1: Overview of all the regret regimes: $\ln T$ (cyan), $\ln(MT)$ (green), $\sqrt{T}$ (yellow), and $T$ (red), depending on the feedback (full or 2-bit) and the assumption on the cdf ($M$-Lipschitz, continuous, or no assumptions).

As in Bolić et al. (2024), we assume that traders' valuations $V, V_1, V_2, \dots$ are generated i.i.d. from an *unknown* distribution $\nu$—a practical assumption for large and stable markets.[2]

Finally, we consider the following two different types of feedback commonly studied in the online learning bilateral trade literature:

- *Full-feedback.* At each round $t$, after having posted the price $P_t$, the broker has access to the traders' valuations $V_{2t-1}$ and $V_{2t}$.
- *2-bit feedback.* At each round $t$, after having posted the price $P_t$, the broker has access to the indicator functions $\mathbb{I}\{V_{2t-1} \le P_t\}$ and $\mathbb{I}\{V_{2t} \le P_t\}$.

The full-feedback model draws its motivation from *direct revelation mechanisms*, where the traders disclose their valuations $V_{2t-1}$ and $V_{2t}$ before each round, but the mechanism has access to this information only after having posted the current bid $P_t$ (Cesa-Bianchi et al., 2021; 2024a).

The 2-bit feedback model corresponds to *posted price* mechanisms, where the broker has access only to the traders' willingness to buy or sell at the proposed posted price, and the valuations $V_{2t-1}$ and $V_{2t}$ are *never* revealed.

### 1.3 Overview of Our Contributions

In the full-feedback case, if the distribution $\nu$ of the traders' valuations has a *continuous* cdf, we design an algorithm (Algorithm 1) suffering $O(\ln T)$ regret in the time horizon $T$ (Theorem 1), and we provide a matching lower bound (Theorem 2). We complement these results by showing that dropping the continuous cdf assumption leads to a worse regret rate of $\Omega(\sqrt{T})$ (Theorem 5), and we design an algorithm (Algorithm 3) achieving $O(\sqrt{T})$ regret (Theorem 6).

In the 2-bit feedback case, if the cdf of the traders' valuations is $M$-Lipschitz, we design an algorithm (Algorithm 2) achieving regret $O\big(\ln(MT)\ln T\big)$ (Theorem 3) where $T$ is the time horizon, and provide a near-matching lower bound $\Omega\big(\ln(MT)\big)$ (Theorem 4). We complement these results by showing that the problem becomes unlearnable if we drop the Lipschitzness assumption (Theorem 7).

For a full summary of our results, see Table 1.

### 1.4 Techniques and Challenges

Online learning with a continuous action domain and full-feedback is usually tackled by discretizing the action domain and then playing an optimal expert algorithm on the discretization, or by directly running exponential weights algorithms in the continuum (Maillard & Munos, 2010; Krichene et al., 2015; Cesa-Bianchi et al., 2024b). These approaches require that the (expected) reward function is Lipschitz and lead to a regret rate of order $\widetilde{O}(\sqrt{T})$. In contrast, our expected reward function is *not* Lipschitz in general. To overcome this challenge, we leverage the specific structure of the problem by proving Lemma 1, which enables us to design an algorithm that achieves an exponentially better regret rate of $O(\ln T)$ even when the underlying cdf—and hence the associated reward function—is only continuous. Moreover, we establish a matching $\Omega(\ln T)$ lower bound that, surprisingly, applies even when the reward function is Lipschitz, demonstrating that additional Lipschitz regularity beyond continuity does not contribute to faster rates in this setting. This lower bound construction is particularly challenging because the shape of the function $p \mapsto \mathbb{E}\big[G_t(p)\big]$ can only be controlled indirectly through the traders' valuation distribution: to avoid exceedingly complex calculations,

---

[2]For further discussion on this assumption, see Appendix B.

extra care is required in selecting appropriate instances. Even then, we needed a subtle and somewhat intricate Bayesian argument to obtain the lower bound.

In the 2-bit feedback model, we remark that the available feedback is enough to reconstruct *bandit* feedback. Consequently, when the underlying cdf—and hence the expected reward function—is $M$-Lipschitz, a viable approach is to discretize the action space $[0,1]$ with $K$ uniformly spaced points and run an optimal bandit algorithm on the discretization. This approach immediately yields a regret rate of order $O(MT/K + \sqrt{KT})$. This bound leads to a regret of order $O(M^{1/3}T^{2/3})$ by tuning $K \coloneqq \Theta(M^{2/3}T^{1/3})$ when $M$ is known to the learner, or of order $O(MT^{2/3})$ by tuning $K \coloneqq \Theta(T^{2/3})$ when the learner does not possess this knowledge. In contrast, we exploit the extra information provided by the 2-bit feedback and the intuition provided by Lemma 1 to devise a binary search algorithm achieving the exponentially better rate of $O(\ln(MT)\ln T)$, with the additional feature of being oblivious to $M$. Our corresponding lower bound shows that this rate is optimal (up to a $\ln T$ factor), demonstrating through an information-theoretic argument that some sort of binary search is essentially a necessary step for optimal learning.

## 1.5 RELATED WORK

Bilateral trade was originally studied in a one-shot setting where a broker has to devise a mechanism to make a buyer and a seller trade, and classical properties like incentive compatibility, individual rationality, budget balance, and efficiency were investigated. Since the pioneering work of Myerson and Satterthwaite and their celebrated impossibility result (Myerson & Satterthwaite, 1983), the study of bilateral trade has grown significantly, particularly from a game-theoretic and approximation perspective (McAfee, 2008; Colini-Baldeschi et al., 2016; 2017; Blumrosen & Mizrahi, 2016; Brustle et al., 2017; Colini-Baldeschi et al., 2020; Babaioff et al., 2020; Dütting et al., 2021; Deng et al., 2022; Kang et al., 2022; Fei, 2022; Archbold et al., 2023; Xu et al., 2024). For a comprehensive overview, refer to Cesa-Bianchi et al. (2024a).

In recent years, the focus has expanded to include online learning settings for bilateral trade. Given their close relevance to our paper, we concentrate our discussion on these works.

In Cesa-Bianchi et al. (2021); Azar et al. (2022); Cesa-Bianchi et al. (2024a; 2023); Bernasconi et al. (2024); Cesa-Bianchi et al. (2024b), the authors examined bilateral trade problems where the reward function is the *gain from trade* and each trader has a fixed role as either a seller or a buyer.

In Cesa-Bianchi et al. (2021), the authors investigated a scenario where seller and buyer valuations form two distinct i.i.d. sequences. In the full-feedback case, they achieved a regret bound of $\widetilde{O}(\sqrt{T})$, which was later refined to $O(\sqrt{T})$ in Cesa-Bianchi et al. (2024a). They also demonstrated a worst-case regret of $\Omega(\sqrt{T})$ even when sellers' and buyers' valuations are independent of each other and their cdfs are Lipschitz. For the 2-bit feedback scenario under i.i.d. valuations, Cesa-Bianchi et al. (2021) proved that any algorithm must suffer linear regret, even under either the $M$-Lipschitz joint cdf assumption or the traders' valuation independence assumption. However, when both conditions are simultaneously satisfied, they proposed an algorithm achieving a regret rate of $\widetilde{O}(M^{1/3}T^{2/3})$, later refined to $O(M^{1/3}T^{2/3})$ in Cesa-Bianchi et al. (2024a). Cesa-Bianchi et al. (2021) also established a worst-case regret lower bound of $\Omega(T^{2/3})$ in this case, which, however, does not display any dependence on $M$.

Cesa-Bianchi et al. (2021; 2024a) also showed that the adversarial bilateral trade problem is unlearnable even with full-feedback. To achieve learnability beyond the i.i.d. case, Cesa-Bianchi et al. (2023; 2024b) explored weakly budget-balanced mechanisms, allowing the broker to post different selling and buying prices as long as the buyer pays more than what the seller receives. They demonstrated that learning can be achieved using weakly budget-balanced mechanisms in the 2-bit feedback setting at a regret rate of $\widetilde{O}(MT^{3/4})$ when the joint seller/buyer cdf may vary over time but is $M$-Lipschitz. Furthermore, for the same setting, they provided a $\Omega(T^{3/4})$ matching lower bound in the time horizon, even when the process is required to be i.i.d., but their lower bound does not feature any dependence on $M$. Azar et al. (2022) investigated whether learning is possible in the adversarial case by considering $\alpha$-regret, achieving $\widetilde{\Theta}(\sqrt{T})$ bounds for 2-regret in full-feedback, and a $\widetilde{O}(T^{3/4})$ upper bound in 2-bit feedback. Following another direction, Bernasconi et al. (2024)

explored globally budget-balanced mechanisms in the adversarial case, showing a $\Theta(\sqrt{T})$ regret rate in full-feedback and a $\widetilde{O}(T^{3/4})$ rate in the 2-bit feedback case.

The closest to our work is Bolić et al. (2024), where the authors studied the same i.i.d. version of our trading problem with flexible seller and buyer roles, but with the target reward function being the *gain from trade*. Under the $M$-Lipschitz cdf assumption, they obtained tight $\Theta(M \ln T)$ regret in the full-feedback case. Surprisingly, in the same full-feedback case, but using our different reward function, we achieve a $\Theta(\ln T)$ regret rate even when the cdf is only continuous: in our case, the additional Lipschitz regularity does not offer any speedup once the continuity assumption is fulfilled. Furthermore, under the $M$-Lipschitz cdf assumption, Bolić et al. (2024) proved a sharp rate of $\Theta(\sqrt{MT})$ in the 2-bit feedback case. Interestingly, using our different reward function, we achieve an exponentially faster upper bound of $O\big(\ln(MT) \ln T\big)$, which is tight up to a $\ln T$ factor. If the Lipschitz cdf assumption is removed, the learning rate for both our problem and the one in Bolić et al. (2024) degrades to $\Theta(\sqrt{T})$ in the full-feedback case, and the problem becomes unlearnable in the 2-bit feedback case.

## 2  THE MEDIAN LEMMA

In this section, we present the Median Lemma (Lemma 1), a simple but crucial result for what follows, and the key upon which our main algorithms are based. At a high level, Lemma 1 states that a broker who aims at maximizing the number of trades should post prices that are as close as possible to the *median* of the (unknown) traders' valuation distribution $\nu$, and the instantaneous regret which the broker incurs by playing any price is (proportional to) the *square* of the distance between the median and the price, if distances are measured with respect to the pseudo-metric induced by the traders' valuation cdf.

**Lemma 1** (The median lemma). *If the cdf $F$ of $\nu$ is continuous, then, for any $t \in \mathbb{N}$ and any $p \in [0, 1]$,*

$$\mathbb{E}\big[G_t(p)\big] = 2F(p)\big(1 - F(p)\big) \qquad and \qquad \frac{1}{2} - \mathbb{E}\big[G_t(p)\big] = 2\left(\frac{1}{2} - F(p)\right)^2 .$$

*In particular, the function $p \mapsto \mathbb{E}\big[G_t(p)\big]$ is maximized at any point $m \in [0, 1]$ such that $F(m) = \frac{1}{2}$.*

Before presenting the proof of Lemma 1, we just remark that points $m \in [0, 1]$ satisfying $F(m) = 1/2$ do exist by the intermediate value theorem, because $F(0) = 0$, $F(1) = 1$, and $F$ is continuous.

*Proof.* For each $t \in \mathbb{N}$ and each $p \in [0, 1]$, we have that

$$\mathbb{E}\big[G_t(p)\big] = \mathbb{P}\Big[\big\{V_{2t-1} \le p < V_{2t}\big\} \cup \big\{V_{2t} \le p \le V_{2t-1}\big\}\Big]$$
$$= \mathbb{P}\big[V_{2t-1} \le p\big]\mathbb{P}\big[p < V_{2t}\big] + \mathbb{P}\big[V_{2t} \le p\big]\mathbb{P}\big[p \le V_{2t-1}\big] = 2F(p)\big(1 - F(p)\big) ,$$

where the second equality follows from additivity and independence, while in the last equality we leveraged the continuity of $F$ to obtain $\mathbb{P}[p \le V_{2t-1}] = \mathbb{P}[p < V_{2t-1}] = 1 - F(p)$. To conclude, it is enough to note that, for each $p \in [0, 1]$ it holds that $1/4 - F(p)\big(1 - F(p)\big) = \big(1/2 - F(p)\big)^2$.  □

## 3  TRADING VOLUME VS GAIN FROM TRADE

In this section, we leverage Lemma 1 to show with a formal example that, unlike trading-volume maximizing brokers, gain-from-trade maximization brokers can be heavily biased towards small segments of the population and, as a result, end up hurting their own bottom lines.

Assume that the distribution of the traders' valuations $V, V_1, V_2, \ldots$ have common density $f$ defined, for all $x \in [0, 1]$, by $f(x) := \big(\frac{1}{\varepsilon} - 1\big)\mathbb{I}\big\{\frac{1}{2} - \varepsilon \le x \le \frac{1}{2}\big\} + \mathbb{I}\{1 - \varepsilon \le x \le 1\}$, for some $\varepsilon \in \big(0, \frac{1}{2}\big)$.

At a high level, this population of traders is clustered into two segments: a *low*-valuation cluster $L$ that believes that the good on sale has a value slightly smaller than $\frac{1}{2}$ and a *high*-valuation cluster $H$ that believes the value is slightly smaller than 1. If $\varepsilon \approx 0$, the overwhelming majority of the population belongs to the low-valuation cluster $L$. In this case, we will prove that a gain-from-trade maximizing

broker would sacrifice the majority of the population to favor trades that include a trader coming from the (extremely small) high-valuation cluster $H$.

Indeed, by Bolić et al. (2024), a gain-from-trade maximizing broker would post the *expectation* $\mathbb{E}[V] = \frac{1}{2}$. In contrast, by Lemma 1, a trade-volume maximizing broker would post the *median* $m \coloneqq \frac{1}{2} - \frac{\varepsilon}{2} \cdot \frac{1-2\varepsilon}{1-\varepsilon}$ of $V$, which is a value roughly in the middle of the low-valuation cluster $L$.

By posting the expectation, the probability of having a trade is, for all $t \in \mathbb{N}$, $\mathbb{P}[V_{2t-1} \wedge V_{2t} \leq \frac{1}{2} \leq V_{2t-1} \vee V_{2t}] = 2(1-\varepsilon)\varepsilon$, which is close to zero when $\varepsilon \approx 0$.

In contrast, by posting the median, the probability of having a trade is, for all $t \in \mathbb{N}$, $\mathbb{P}[V_{2t-1} \wedge V_{2t} \leq m \leq V_{2t-1} \vee V_{2t}] = \frac{1}{2}$, which is always bounded away from zero, irrespectively of $\varepsilon$.

This shows two ways in which (unlike a trade-volume maximizing broker) a gain-from-trade maximizing broker is biased towards favoring the high valuation cluster $H$. First, they are willing to accept that only a negligible fraction of the population will trade. Second, being $\mathbb{E}[V] = \frac{1}{2}$, they only (with probability 1) allow trades where one of the two traders comes from the high-valuation cluster $H$, resulting in *only* the high-valuation trader making a large profit, while the low valuation trader is left with a profit of order $\varepsilon \approx 0$, even in the low-probability event where they are given the opportunity to trade. It is easy to imagine that, in real life, such a bias would cause the low-valuation traders in $L$ to leave the broker, in turn greatly reducing the broker's own profit.

For further discussion on the trading-volume metric, see Appendix A.

## 4 FULL-FEEDBACK

We now investigate how the broker should behave to maximize the number of trades in the full-feedback case where after each interaction the traders' valuations are disclosed. We begin by studying the full-feedback case under the continuous cdf assumption. In this case, taking inspiration from Lemma 1, a natural strategy is to play the *empirical median*, which leads to Algorithm 1.

---

**Algorithm 1:** Follow the Empirical Median (FEM)

Post $P_1 \coloneqq 1/2$ and receive feedback $V_1, V_2$;
**for** *time* $t = 2, 3, \ldots$ **do**

    Post the empirical median $P_t \coloneqq \frac{1}{2}\left(V_{2(t-1)}^{(t-1)} + V_{2(t-1)}^{(t)}\right)$, where $V_{2(t-1)}^{(1)}, \ldots, V_{2(t-1)}^{(2(t-1))}$ are the order statistics of the observed sample $V_1, \ldots, V_{2(t-1)}$, and receive feedback $V_{2t-1}, V_{2t}$;

---

The next theorem leverages Lemma 1 to show that Algorithm 1 suffers regret $O(\ln T)$ when the traders' valuation cdf is continuous.

**Theorem 1.** *If $\nu$ has a continuous cdf $F$, the regret of FEM satisfies, for all time horizons $T \in \mathbb{N}$,*

$$R_T \leq \frac{1}{2} + \frac{\pi}{2}\big(1 + \ln(T-1)\big).$$

*Proof.* Without loss of generality, we can (and do!) assume that $T \geq 2$. Then, we have

$$R_T \leq \frac{1}{2} + \max_{p \in [0,1]} \mathbb{E}\left[\sum_{t=2}^{T} G_t(p)\right] - \mathbb{E}\left[\sum_{t=2}^{T} G_t(P_t)\right] = \frac{1}{2} + 2 \cdot \sum_{t=2}^{T} \mathbb{E}\left[\left(\frac{1}{2} - F(P_t)\right)^2\right]$$

Now, let $m \in [0,1]$ be such that $F(m) = 1/2$, and let $V$ be a random variable whose distribution is $\nu$, independent of $V_1, V_2, \ldots$. Then, for any $t \in \mathbb{N}$ such that $t \geq 2$ we have

$$\mathbb{E}\left[\left(\frac{1}{2} - F(P_t)\right)^2\right] = \mathbb{E}\left[\left(\mathbb{P}[m \leq V \leq P_t \mid P_t]\right)^2\right] + \mathbb{E}\left[\left(\mathbb{P}[P_t \leq V \leq m \mid P_t]\right)^2\right] =: (I) + (II).$$

Now, for the term $(I)$, leveraging the fact that $V$ and $P_t$ are independent of each other, together with the Minkowski's integral inequality (see, e.g., (Stein, 1970, Appendix A.1)), we have:

$$\sqrt{(I)} = \sqrt{\mathbb{E}\left[\left(\mathbb{E}\left[\mathbb{I}\{m \le V \le P_t\} \mid P_t\right]\right)^2\right]} \le \mathbb{E}\left[\sqrt{\mathbb{E}\left[\left(\mathbb{I}\{m \le V \le P_t\}\right)^2 \mid V\right]}\right]$$

$$= \mathbb{E}\left[\sqrt{\mathbb{P}[m \le V \le P_t \mid V]}\right] = \int_{[m,1]} \sqrt{\mathbb{P}[x \le P_t]}\, \mathrm{d}\mathbb{P}_V(x) = \int_{[m,1]} \sqrt{\mathbb{P}[x \le P_t]}\, \mathrm{d}\nu(x) = (\star)$$

For each $x \in [0,1]$ and for any $s \in \mathbb{N}$, let $B_s(x) := \mathbb{I}\{x \le V_s\}$, and notice that $B_1(x), B_2(x), \dots$ is an i.i.d. sequence of Bernoulli random variables of parameter $1 - F(x)$. Let $V_{2(t-1)}^{(1)}, \dots, V_{2(t-1)}^{(2(t-1))}$ be the order statistics of the observed sample $V_1, \dots, V_{2(t-1)}$. For any $x \in [m, 1]$, observing that $F(x) - \frac{1}{2} \ge 0$ and $\mathbb{P}[x \le P_t] \le \mathbb{P}\left[x \le V_{2(t-1)}^{(t)}\right] \le \mathbb{P}\left[\sum_{s=1}^{2(t-1)} B_s(x) \ge t - 1\right]$, we can leverage Hoeffding's inequality to obtain

$$\mathbb{P}[x \le P_t] \le \mathbb{P}\left[\sum_{s=1}^{2(t-1)} B_s(x) \ge t - 1\right] = \mathbb{P}\left[\sum_{s=1}^{2(t-1)} \frac{B_s(x)}{2(t-1)} - (1 - F(x)) \ge \frac{t-1}{2(t-1)} - (1 - F(x))\right]$$

$$= \mathbb{P}\left[\sum_{s=1}^{2(t-1)} \frac{B_s(x)}{2(t-1)} - (1 - F(x)) \ge F(x) - \frac{1}{2}\right] \le e^{-4(t-1)\left(F(x) - \frac{1}{2}\right)^2} = e^{-4(t-1)\left(\nu[[0,x]] - \frac{1}{2}\right)^2},$$

from which, by the change of variable formula (Revuz & Yor, 2013, Proposition 4.10, Chapter 1), it follows also that

$$(\star) \le \int_{[m,1]} \sqrt{\exp\left(-4(t-1)\left(\nu[[0,x]] - \frac{1}{2}\right)^2\right)}\, \mathrm{d}\nu(x) = \int_{1/2}^{1} \exp\left(-2(t-1)\left(\frac{1}{2} - u\right)^2\right) \mathrm{d}u$$

$$\le \frac{1}{\sqrt{2(t-1)}} \int_0^\infty \exp\left(-r^2\right) \mathrm{d}r = \frac{\sqrt{\pi}}{2\sqrt{2}} \cdot \frac{1}{\sqrt{t-1}},$$

and hence $(I) \le \frac{\pi}{8(t-1)}$. Analogously, we can prove that $(II) \le \frac{\pi}{8(t-1)}$. Hence,

$$R_T \le \frac{1}{2} + \frac{\pi}{2} \cdot \sum_{t=2}^{T} \frac{1}{t-1} = \frac{1}{2} + \frac{\pi}{2} + \frac{\pi}{2} \cdot \sum_{t=2}^{T-1} \int_{t-1}^{t} \frac{1}{t}\, \mathrm{d}s \le \frac{1}{2} + \frac{\pi}{2} + \frac{\pi}{2} \cdot \int_{1}^{T-1} \frac{1}{s}\, \mathrm{d}s = \frac{1}{2} + \frac{\pi}{2}\left(1 + \ln(T-1)\right). \quad \square$$

We now establish the optimality of FEM by demonstrating a matching $\Omega(\ln T)$ regret lower bound. We remark that this result holds even when competing against underlying distributions that have a 2-Lipschitz cdf, thus proving the optimality of FEM even under the Lipschitz cdf assumption.

**Theorem 2.** *There exist two numerical constants $c_1$ and $c_2$ such that, for any time horizon $T \ge c_2$, the worst-case regret of any full-feedback algorithm satisfies*

$$\sup_{\nu \in \mathcal{D}_2} R_T^\nu \ge c_1 \ln T,$$

*where $R_T^\nu$ is the regret at time $T$ of the algorithm when the i.i.d. sequence of traders' valuations follows the distribution $\nu$, and $\mathcal{D}_2$ is the set of all distributions $\nu$ that admit a 2-Lipschitz cdf.*

Due to space constraints, we defer the (long and technical) proof of this result to Appendix C and only present a short, high-level sketch here.

*Proof sketch.* In the proof, we build a family of 2-Lipschitz cdfs $F_\varepsilon$ parameterized by $\varepsilon \in [0,1]$, so that if two instances are parameterized by $\varepsilon$ and $\varepsilon'$ respectively, then their medians are $\Theta\left(|\varepsilon - \varepsilon'|\right)$-away from each other (Figure 1). The high-level idea is to leverage a Bayesian argument to show that if the underlying instance $F_E$ is such that $E$ is drawn uniformly at random in $[0,1]$, then, at round $t$, the broker cannot reliably determine prices that are much closer than $1/\sqrt{t}$ to the corresponding median $m_E$ when distances are measured with respect to the metric induced by the cdf $F_E$. This, together with our key Lemma 1, leads to the conclusion. $\square$

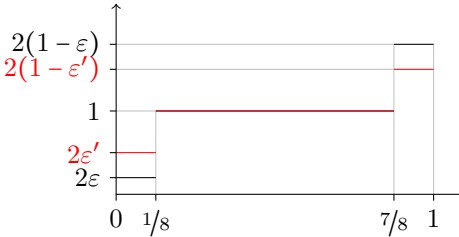 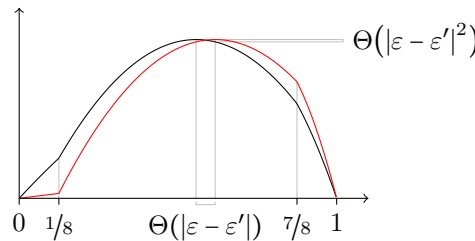

Figure 1: Qualitative plots of the densities $f_\varepsilon$, $f_{\varepsilon'}$ (left) and corresponding expected rewards (right) used in the proof of Theorem 2 for two values $\varepsilon, \varepsilon' > 0$. Note that the difference in reward by posting a price that is optimal for one instance $\varepsilon'$ when the actual instance is $\varepsilon$ is $\Theta\big(|\varepsilon - \varepsilon'|^2\big)$.

## 5   2-BIT FEEDBACK

We start the study of the 2-bit feedback case under the assumption that the traders' valuation distribution admits a Lipschitz cdf $F$. The algorithm we propose (Algorithm 2) is based on the following observation: by posting any price $p$, the broker has access to two noisy realizations of $F(p)$. Recalling that Lemma 1 suggests tracking the median of $F$ (i.e., a point $m$ at which $F(m) = 1/2$), and since $F$ is a non-decreasing function, we can proceed using a natural binary search strategy to move toward the median. This can be done in epochs: in each one, we repeatedly test a (dyadic) price until the first time we can confidently decide that the median is to the left or right of the current price.

---

**Algorithm 2:** Median Binary Search (MBS)

---

**Input:** Confidence parameter $\delta \in (0, 1)$, time horizon $n \in \mathbb{N}$;
**Initialization:** $Q_1 \coloneqq \frac{1}{2}$, $\tau \coloneqq 1$, $t \coloneqq 1$;
**while** *time $t \le n$* **do**
$\quad$ Let $s \coloneqq 0$, $Y_{\tau,s} \coloneqq 0$, $t_{\tau-1} \coloneqq t - 1$;
$\quad$ **while** *time $t \le n$* **do**
$\quad\quad$ Post $P_t \coloneqq Q_\tau$ and receive feedback $\mathbb{I}\{V_{2t-1} \le P_t\}$, $\mathbb{I}\{V_{2t} \le P_t\}$;
$\quad\quad$ Update $s \coloneqq s + 2$, $Y_{\tau,s} \coloneqq Y_{\tau,s-2} + \mathbb{I}\{V_{2t-1} \le P_t\} + \mathbb{I}\{V_{2t} \le P_t\}$, $t \coloneqq t + 1$;
$\quad\quad$ **if** $\frac{1}{s}Y_{\tau,s} + \sqrt{\frac{\ln(2/\delta)}{2s}} < \frac{1}{2}$ **then** let $Q_{\tau+1} \coloneqq Q_{\tau+1} + \frac{1}{2^{\tau+1}}$, $s_\tau \coloneqq s$, $\tau \coloneqq \tau + 1$, and **break**;
$\quad\quad$ **else if** $\frac{1}{s}Y_{\tau,s} - \sqrt{\frac{\ln(2/\delta)}{2s}} > \frac{1}{2}$ **then** let $Q_{\tau+1} \coloneqq Q_{\tau+1} - \frac{1}{2^{\tau+1}}$, $s_\tau \coloneqq s$, $\tau \coloneqq \tau + 1$, and **break**;

---

We now show that a suitably tuned Algorithm 2 has regret guarantees of $O\big(\ln(MT)\ln(T)\big)$. In particular, we stress that the tuning of Algorithm 2 does not need prior knowledge of $M$. Due to space constraints, we defer the full proof of the next result to Appendix D.

**Theorem 3.** *If $\nu$ has an $M$-Lipschitz cdf $F$ for some $M \ge 1$, then, for all time horizons $T \in \mathbb{N}$, the regret of MBS tuned with parameters $\delta \coloneqq 2/T^3$ and $n \coloneqq T$ satisfies*

$$R_T \le 2 + 6\log_2(MT)\ln(T) .$$

*Proof sketch.* The proof is based on the following observations. First, during an epoch where a price $p$ is tested, given that one has to distinguish if the parameter $F(p)$ of a sequence of Bernoulli random variables is bigger or smaller than $1/2$, a concentration argument shows that the duration of this epoch is at most $O\big(\ln(1/\delta)/\big(1/2 - F(p)\big)^2\big)$, where $\delta$ is the confidence parameter. Second, by Lemma 1, the broker regrets $2\big(1/2 - F(p)\big)^2$ by playing a price $p$, and hence the total regret of an epoch where the broker tests $p$ is at most $O\big(\ln(1/\delta)\big)$. We then use the fact that the $F$ is $M$-Lipschitz to prove that, after at most $O\big(\log_2(MT)\big)$ epochs, the cumulative regret that the algorithm suffers from that point onward is constant, and conclude by showing that the tuning of $\delta$ leads to the stated guarantees. $\qquad\square$

We now show that Algorithm 2 is optimal, up to a multiplicative $\ln T$ term. Due to space constraints, we defer the full proof of this result to Appendix E.

**Theorem 4.** *There exist two numerical constants $c_1$ and $c_2$ such that for any $M \geq 16$ and any time horizon $T \geq c_2 \log_2(M)$, the worst-case regret of any 2-bit feedback algorithm satisfies*

$$\sup_{\nu \in \mathcal{D}_M} R_T^\nu \geq c_1 \ln(MT) \,,$$

*where $R_T^\nu$ is the regret at time $T$ of the algorithm when the i.i.d. sequence of traders' valuations follows the distribution $\nu$, and $\mathcal{D}_M$ is the set of all distributions $\nu$ that admits an $M$-Lipschitz cdf.*

*Proof sketch.* The proof builds a family of distributions, each supported in a different region of length $\Theta(1/M)$, whose cdfs are $M$-Lipschitz. To avoid suffering linear regret if the traders' valuations are generated according to one of these distributions, the broker has to detect the corresponding support. To accomplish this task, we show that the broker is essentially forced to solve a binary search problem that needs $\log_2(M)$ rounds in each of which the instantaneous regret is constant. Noticing that any regret lower bound for full-feedback algorithms also applies to 2-bit feedback algorithms, the $\ln T$ lower bound of Theorem 2 together with the binary search $\ln M$ lower bound yield a lower bound of $\Omega\big(\max(\ln T, \ln M)\big) = \Omega\big(\ln(MT)\big)$. $\qquad\square$

## 6    Non-Lipschitz or Discontinuous Pdfs

We now investigate how the problem changes if we lift the assumption that $\nu$ has a Lipschitz or continuous cdf. First, note that when the cdf of $\nu$ is not continuous, Lemma 1, and, consequently, the guarantees of Theorem 1, no longer hold. Indeed, in general, no full-feedback algorithm can achieve regret guarantees better than $\sqrt{T}$. As shown in the proof of the next theorem, the reason is that our problem contains instances that resemble online learning with expert advice (with 2 experts), which has a known lower bound of $\Omega(\sqrt{T})$.

**Theorem 5.** *There exist two numerical constants $c_1$ and $c_2$ such that, for any time horizon $T \geq c_2$, the worst-case regret of any full feedback algorithm satisfies*

$$\sup_{\nu \in \mathcal{D}} R_T^\nu \geq c_1 \sqrt{T} \,,$$

*where $R_T^\nu$ is the regret at time $T$ of the algorithm when the i.i.d. sequence of traders' valuations follows the distribution $\nu$, and $\mathcal{D}$ is the set of all distributions $\nu$.*

*Proof sketch.* For each $\varepsilon \in [-1/4, 1/4]$, define $\nu_\varepsilon \coloneqq \frac{1-\varepsilon}{4}\delta_0 + \frac{1}{4}\delta_{1/3} + \frac{1}{4}\delta_{2/3} + \frac{1+\varepsilon}{4}\delta_1$, where, for any $a \in \mathbb{R}$, we denoted by $\delta_a$ the Dirac's delta probability measure centered at $a$. Let $(V_{\varepsilon,t})_{\varepsilon \in [-1/4,1/4], t \in \mathbb{N}}$ be an independent family such that for each $\varepsilon \in [-1/4, 1/4]$ the sequence $V_{\varepsilon,1}, V_{\varepsilon,2}, \ldots$ is i.i.d. with common distribution $\nu_\varepsilon$. For each $\varepsilon \in [-1/4, 1/4]$, each $t \in \mathbb{N}$, and each $p \in [0, 1]$, define $\mathrm{G}_{\varepsilon,t}(p) \coloneqq g(p, V_{\varepsilon,2t-1}, V_{\varepsilon,2t})$. Straightforward computations show that, for each $\varepsilon \in [-1/4, 1/4]$ and each $t \in \mathbb{N}$, the function $p \mapsto \mathbb{E}\big[\mathrm{G}_{\varepsilon,t}(p)\big]$ is maximized at $1/3$ or at $2/3$, with any other point having an expected reward that is less than $31/256$-away from the minimum expected reward achieved at $1/3$ or $2/3$. Furthermore, for any $\varepsilon \in [-1/4, 1/4]$ and any $t \in \mathbb{N}$, the maximum is at $1/3$ or $2/3$ depending on whether $\varepsilon < 0$ or $\varepsilon > 0$, given that $\mathbb{E}\big[\mathrm{G}_{\varepsilon,t}(1/3)\big] = \frac{11}{16} - \frac{\varepsilon}{8} - \frac{\varepsilon^2}{8}$ and $\mathbb{E}\big[\mathrm{G}_{\varepsilon,t}(2/3)\big] = \frac{11}{16} + \frac{\varepsilon}{8} - \frac{\varepsilon^2}{8}$, from which it follows also that $\mathbb{E}\big[\mathrm{G}_{\varepsilon,t}(2/3)\big] - \mathbb{E}\big[\mathrm{G}_{\varepsilon,t}(1/3)\big] = \frac{\varepsilon}{4}$. Hence, in order not to suffer $\Omega\big(|\varepsilon| T\big)$ regret, an algorithm has to detect the *sign* of $\varepsilon$. However, a standard information-theoretic argument shows that a sample of order $\Omega(1/\varepsilon^2)$ is required in order to detect the sign of $\varepsilon$. During this period, the best any algorithm can do is to play blindly in the set $\{1/3, 2/3\}$, incurring in a cumulative regret of order $\Omega\left(\frac{1}{\varepsilon^2} \cdot |\varepsilon|\right) = \Omega\left(1/|\varepsilon|\right)$. Overall, any learner has to suffer $\Omega\left(\min\left(\frac{1}{|\varepsilon|}, |\varepsilon| T\right)\right)$ worst-case regret, which, by tuning $|\varepsilon| = \Theta(1/\sqrt{T})$, leads to a worst-case regret lower bound of $\Omega(\sqrt{T})$. $\qquad\square$

We now focus on the upper bound. A closer look at the proof of Lemma 1 shows that if we drop the cdf continuity assumption in the Median Lemma, the formula generalizes to

$$\mathbb{E}\big[\mathrm{G}_t(p)\big] = 2F(p)\big(1 - F(p)\big) + F(p)F^\circ(p) =: \Psi(p) \,, \qquad \forall p \in [0, 1], \; \forall t \in \mathbb{N},$$

with no assumptions on $\nu$, and where $F$ is the cdf of $\nu$ and we defined $F^\circ(p) \coloneqq \nu\big[\{p\}\big]$. This suggests the strategy of building an empirical proxy $\hat{\Psi}_t$ of $\Psi$ with the feedback available at time $t$, and posting

prices that maximize $\hat{\Psi}_t$. By replacing the theoretical quantities by their empirical counterparts, for any $t \in \mathbb{N}$ and any $p \in [0,1]$, we define an empirical proxy for $\Psi(p)$ as follows:

$$\hat{\Psi}_{t+1}(p) \coloneqq 2\frac{1}{2t}\sum_{s=1}^{2t}\mathbb{I}\{V_s \le p\}\frac{1}{2t}\sum_{s=1}^{2t}\mathbb{I}\{p < V_s\} + \frac{1}{2t}\sum_{s=1}^{2t}\mathbb{I}\{V_s \le p\}\frac{1}{2t}\sum_{s=1}^{2t}\mathbb{I}\{V_s = p\} \ .$$

This definition leads to Algorithm 3.

---

**Algorithm 3:** Follow the Empirical $\Psi$ (FE$\Psi$)

---

Post $P_1 = \frac{1}{2}$ and receive feedback $V_1, V_2$;
**for** *time* $t = 2, 3, \ldots$ **do**
    Post $P_t \in \operatorname{argmax}_{p \in [0,1]} \hat{\Psi}_t(p)$ and receive feedback $V_{2t-1}, V_{2t}$;

---

We now state regret guarantees for Algorithm 3. The proof of the following result (which hinges on showing that $\hat{\Psi}_t$ is *uniformly* close to $\Psi$ with high probability) is deferred to Appendix F.

**Theorem 6.** *For all time horizons $T \in \mathbb{N}$, the regret of FE$\Psi$ satisfies*

$$R_T \le 1 + 8\sqrt{\pi} \cdot \sqrt{T-1} \ .$$

We conclude by showing that, without the Lipschitz cdf assumption, the 2-bit feedback problem is unlearnable. This can be deduced as a simple corollary of the proof of Theorem 4. Specifically, we can obtain a linear worst-case lower bound for any 2-bit feedback algorithm, even if we assume that the underlying distribution has a continuous cdf.

**Theorem 7.** *There exist two numerical constants $c_1$ and $c_2$ such that, for any time horizon $T \ge c_2$, the worst-case regret of any 2-bit feedback algorithm satisfies*

$$\sup_{\nu \in \mathcal{D}_c} R_T^\nu \ge c_1 T \ ,$$

*where $R_T^\nu$ is the regret at time $T$ of the algorithm when the i.i.d. sequence of traders' valuations follows the distribution $\nu$, and $\mathcal{D}_c$ is the set of all distributions $\nu$ that admits a continuous cdf.*

*Proof.* As a consequence of the last part of the proof of Theorem 4 (see Appendix E) we have that, for any time horizon $T \ge 4$, if we set $M \coloneqq 2^T$, then the conditions $M \ge 16$ and $T \ge \log_2(M)$ in that proof holds, and hence, any 2-bit feedback algorithm has worst-case regret that is at least $\frac{1}{4\ln 2}\ln M = \frac{1}{4\ln 2}T$. $\square$

## 7    CONCLUSIONS AND OPEN PROBLEMS

Motivated by maximizing trading volume in OTC markets, we proposed a novel objective that departs from the classical *gain-from-trade* reward studied in the bilateral trade literature. For this new problem, we investigated optimal brokerage strategies from an online learning perspective. Under the assumption that traders are free to sell or buy depending on the trading price and that traders' valuations form an i.i.d. sequence, we provided a complete picture with matching (up to, at most, logarithmic factors) upper and lower bounds in all the proposed settings, fleshing out the role of regularity assumptions in achieving these fast regret rates.

In addition to closing the logarithmic $\ln T$ gap in the regret rate of the 2-bit feedback setting, a few other future research directions are to find non-stationary variants of this problem where learning is still achievable, investigate trading volume maximization when traders have definite seller and buyer roles, and explore the contextual version of the problem when the broker has access to relevant side information before posting each price. One could also consider the 1-*bit* feedback setting, where the learner receives no individual feedback from traders but only observes if trades occurred or not. This is a desirable property of the mechanism as it ensures complete privacy regarding the preferences of the traders. Lastly, a natural question arising from our unlearnability result in the 2-bit feedback setting is if learning is possible when the objective is weakened to $\alpha$-regret. We leave the investigation of these intriguing problems to future research.

ACKNOWLEDGMENTS

TC gratefully acknowledges the support of the University of Ottawa through grant GR002837 (Start-Up Funds) and that of the Natural Sciences and Engineering Research Council of Canada (NSERC) through grants RGPIN-2023-03688 (Discovery Grants Program) and DGECR2023-00208 (Discovery Grants Program, DGECR - Discovery Launch Supplement). RC is partially supported by the MUR PRIN grant 2022EKNE5K (Learning in Markets and Society), the FAIR (Future Artificial Intelligence Research) project, funded by the NextGenerationEU program within the PNRR-PE-AI scheme, the EU Horizon CL4-2022-HUMAN-02 research and innovation action under grant agreement 101120237, project ELIAS (European Lighthouse of AI for Sustainability).

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

## A    FURTHER DISCUSSION ON THE TRADING-VOLUME METRIC

To further elaborate on our choice of maximizing the trading volume, we first compute the individual expected utilities of the traders in our motivating example in Section 3, in the two cases in which the learner posts the price $1/2$ maximizing the gain from the trade and the price $m = \frac{1}{2} - \frac{\varepsilon}{2}\frac{1-2\varepsilon}{1-\varepsilon}$ maximizing the trading volume. A direct verification shows the following:

1. The selling expected utility of traders at $1/2$ is $\varepsilon^2 - \varepsilon^3 = \Theta(\varepsilon^2)$.
2. The buying expected utility of traders at $1/2$ is $\varepsilon - 2\varepsilon^2 + \varepsilon^3 = \Theta(\varepsilon)$.
3. Thus, the gain from trade at $1/2$ is $\varepsilon - 2\varepsilon^2 = \Theta(\varepsilon)$.

On the other hand:

1. The selling expected utility of traders at $m$ is $\frac{1}{8}\varepsilon + \frac{1}{4}\varepsilon^2 = \Theta(\varepsilon)$.
2. The buying expected utility of traders at $m$ is $\frac{5}{8}\varepsilon = \Theta(\varepsilon)$.
3. Thus, the gain from trade at $m$ is $\frac{3}{4}\varepsilon + \frac{1}{4}\varepsilon^2 = \Theta(\varepsilon)$.

Since the distribution of the traders' valuations is continuous and supported on the union $\left[\frac{1}{2} - \varepsilon, \frac{1}{2}\right] \cup [1 - \varepsilon, 1]$ of a low and a high-valuation cluster (with the overwhelming majority of the population concentrated in the low-valuation cluster), then, posting the price $1/2$ that maximizes the gain from trade has the following consequences:

- Traders from the high-valuation cluster (who enter the market with probability $\Theta(\varepsilon)$) will trade with probability $\Theta(1)$ and earn utility $\Theta(1)$.
- Traders from the low-valuation cluster (who enter the market with probability $\Theta(1)$) trade with probability $\Theta(\varepsilon)$ and earn utility $\Theta(\varepsilon)$.

This is arguably an unfair division of utilities because traders in the low-valuation cluster (which constitutes the near-totality of the population) are effectively made unable to trade with each other and, as a consequence, end up earning negligible $\Theta(\varepsilon^2)$ expected utilities. In this case, the near-totality of the gain from trade comes from the $\Theta(\varepsilon)$ expected utility accrued by the high-valuation traders.

In contrast, posting the price $m$ that maximizes the trading volume has the following consequences.

- Traders from the high-valuation cluster (who enter the market with probability $\Theta(\varepsilon)$) will trade with probability $\Theta(1)$ and earn utility $\Theta(1)$.
- Traders from the low-valuation cluster (who enter the market with probability $\Theta(1)$) trade with probability $\Theta(1)$ and earn utility $\Theta(\varepsilon)$.

In this case, low-valuation traders can trade with each other (as well as with high-valuation traders), and consequently, their expected utilities rise to $\Theta(\varepsilon)$. Both low and high-valuation traders accrue $\Theta(\varepsilon)$ expected utilities, maintaining in particular the same order $\Theta(\varepsilon)$ of expected gain from trade.

However, there is no free lunch. The fairness obtained by moving away from directly maximizing the gain from trade comes at the cost that, in general, the gain from trade won't be *exactly* maximized. Take the previous instance as an example. The trading-volume objective effectively manages to drive a balance of the utilities of traders coming from high and low-valuation clusters of the population but at the price of slightly lowering the gain from trade (note that although the order is the same, the

gain from trade at $m$ is lower than the optimal one by a multiplicative constant). If $\varepsilon$ is a constant (however small) independent of the time horizon, this will result in a linear regret with respect to the gain from trade.

This should not come as a surprise, though, as it is what one should expect from *fair* objectives: generally speaking, they are not as efficient as *unfair* ones. This is a point worth raising in a broader sense. Is it worth losing a fraction of the global value to obtain a fairer distribution of a slightly lower total wealth? Perhaps. This is a profound philosophical question that reaches far beyond the scope of our work and whose answer probably varies depending on the circumstances. Still, we believe it is valuable to investigate both options and, at the very least, shine a light on the possible pros and cons of the two, as well as to flesh out the different techniques that are needed to tackle each variant of the problem.

## B  FURTHER DISCUSSION ON THE I.I.D. ASSUMPTION

**Same-distribution assumption.**    Unlike bilateral trade settings where traders join the market as either buyers or sellers (in which case, it is natural to assume that the populations of buyers and sellers have distinct valuation distributions), in our setting, each trader has no predetermined intent to buy or sell. Instead, they simply hold a valuation for the item on sale and are willing to either buy or sell depending on the price proposed by the broker.

Settings with flexible seller/buyer roles have already appeared in the literature and are prevalent in OTC markets. Think of the stock market, or markets of gems and minerals, but also (perhaps less obviously) agricultural markets, as described in (Sherstyuk et al., 2020): "*Consider, for example, markets for agricultural goods [...]. Participants in agricultural markets [...] may switch between seller and buyer roles depending on individual outcomes, consumption needs, and trading opportunities. [...] A considerable literature in development economics studies agricultural households navigating such decisions (e.g., Singh et al. (1986), Key et al. (2000), Taylor and Adelman (2003), and Barrett (2008)).*"

In all these markets, there are no two distinct populations of buyers and sellers. There is simply one population of *traders*.

This naturally leads to the assumption that the traders' valuations come from a distribution modeling fluctuation of the market value of the item on sale.

To elaborate further, the reader might wonder: "What if there are multiple segments of the population, each with its own skewed idea of the market price of the item?" If at every round, two (possibly different) population segments are drawn i.i.d. and then the valuations of the two traders are drawn i.i.d. from the (possibly different) valuations distribution of each segment, would our techniques still apply? The answer is yes. To see it, simply note that the setting we investigate is context-free, i.e., we do not assume that any information about the traders that come each round is available to the learner before they propose the current trading price at each round. Without knowing a priori the segment each trader belongs to, writing the expected utility explicitly shows that the problem becomes equivalent to our formulation, where the single distribution is the mixture of all the distributions coming from different segments.

Therefore, the same-distribution assumption in a context-free setting where traders are allowed to buy or sell depending on current market conditions is essentially done without loss of generality.

In a contextual setting, instead, things change, and different techniques are needed depending on the assumptions relating contextual information to market prices. Recent attempts at investigating contextual models for the brokerage problem can be found in (Bachoc et al., 2024b; 2025).

**Incentive-compatibility and the i.i.d. assumption.**    Consistent with the broader body of bilateral trade literature studied from an online learning perspective, our mechanism is truthful for traders who make "take-it-or-leave-it" decisions and subsequently leave the broker permanently. In such scenarios, traders have no incentive to misreport their valuations, as doing so would yield no benefit and could instead result in forfeiting the opportunity to secure a non-negative utility. This assumption aligns naturally with the i.i.d. framework, which can be interpreted as representing traders drawn

## C   PROOF OF THEOREM 2

For each $\varepsilon \in [0,1]$, consider the following density function (see Figure 1, left)

$$f_\varepsilon \colon [0,1] \to [0,2], \qquad x \mapsto 2\varepsilon \mathbb{I}\left\{x \le \frac{1}{8}\right\} + \mathbb{I}\left\{\frac{1}{8} < x < \frac{7}{8}\right\} + 2(1-\varepsilon)\mathbb{I}\left\{x \ge \frac{7}{8}\right\},$$

Notice that, for each $\varepsilon \in [0,1]$ the cumulative function associated to the density $f_\varepsilon$ is 2-Lipschitz with explicit expression given by

$$F_\varepsilon \colon [0,1] \to [0,1], \quad x \mapsto 2\varepsilon x \mathbb{I}\left\{x \le \frac{1}{8}\right\} + \left(\frac{2\varepsilon - 1}{8} + x\right)\mathbb{I}\left\{\frac{1}{8} < x < \frac{7}{8}\right\} + \left(2\varepsilon - 1 - 2(\varepsilon-1)x\right)\mathbb{I}\left\{x \ge \frac{7}{8}\right\}.$$

Consider for each $\varepsilon \in [0,1]$, an i.i.d. sequence $(B_{\varepsilon,t})_{t\in\mathbb{N}}$ of Bernoulli random variables of parameter $\varepsilon$, an i.i.d. sequence $(D_t)_{t\in\mathbb{N}}$ of Bernoulli random variables of parameter $\frac{1}{4}$, an i.i.d. sequence $(U_t)_{t\in\mathbb{N}}$ of uniform random variables on $[0,1]$, and a uniform random variable $E$ on $[0,1]$, such that $\big((B_{\varepsilon,t})_{t\in\mathbb{N},\varepsilon\in[0,1]}, (D_t)_{t\in\mathbb{N}}, (U_t)_{t\in\mathbb{N}}, E\big)$ is an independent family. For each $\varepsilon \in [0,1]$ and $t \in \mathbb{N}$, define

$$V_{\varepsilon,t} \coloneqq D_t \cdot \left(B_{\varepsilon,t}\frac{U_t}{8} + (1-B_{\varepsilon,t})\frac{7+U_t}{8}\right) + (1-D_t)\cdot\left(\frac{1}{8} + \frac{3}{4}U_t\right). \tag{1}$$

Tedious but straightforward computations show that, for each $\varepsilon \in [0,1]$ the sequence $(V_{\varepsilon,t})_{t\in\mathbb{N}}$ is i.i.d. with common density given by $f_\varepsilon$, and this sequence is independent of $E$. For any $\varepsilon \in [0,1]$, $p \in [0,1]$, and $t \in \mathbb{N}$, let $\mathrm{G}_{\varepsilon,t}(p) \coloneqq g(p, V_{\varepsilon,2t-1}, V_{\varepsilon,2t})$ (for a qualitative representation of its expectation, see Figure 1, right). We now show how to lower bound the worst-case regret of any arbitrary deterministic algorithm for the full-feedback setting $(\alpha_t)_{t\in\mathbb{N}}$, i.e., a sequence of functions $\alpha_t \colon \big([0,1] \times [0,1]\big)^{t-1} \to [0,1]$ where each element maps past feedback into a price (with the convention that $\alpha_1$ is a number in $[0,1]$). We remark that we do not lose any generality in considering only deterministic algorithms given that we are in a stochastic i.i.d. setting, and the minimax regret over deterministic algorithms coincides with that over randomized algorithms. For each $t \in \mathbb{N}$, define $\widetilde{\alpha}_t \colon \big([0,1] \times [0,1]\big)^{t-1} \to \left[\frac{1}{8}, \frac{7}{8}\right]$ equal to $\alpha_t$ whenever $\alpha_t$ takes values in $\left[\frac{1}{8}, \frac{7}{8}\right]$, and equal to $1/2$ otherwise. Notice that for each $\varepsilon \in [0,1]$ it holds that $(F_\varepsilon \circ \widetilde{\alpha}_t)\cdot(1 - F_\varepsilon \circ \widetilde{\alpha}_t) \ge (F_\varepsilon \circ \alpha_t)\cdot(1 - F_\varepsilon \circ \alpha_t)$, and hence, due to Lemma 1, for each $t \in \mathbb{N}$, it holds that $\mathbb{E}\big[\mathrm{G}_{\varepsilon,t}\big(\widetilde{\alpha}_t(V_{\varepsilon,1}, \ldots, V_{\varepsilon,2(t-1)})\big)\big] \ge \mathbb{E}\big[\mathrm{G}_{\varepsilon,t}\big(\alpha_t(V_{\varepsilon,1}, \ldots, V_{\varepsilon,2(t-1)})\big)\big]$. Notice also that for each $\varepsilon \in [0,1]$, we have that $m_\varepsilon \coloneqq \frac{5-2\varepsilon}{8}$ is the unique element in $[0,1]$ such that $F_\varepsilon(m_\varepsilon) = 1/2$. For any time horizon $T \ge 144$, we have that the worst-case regret of the algorithm $(\alpha_t)_{t\in\mathbb{N}}$ can be lower bounded as follows

$$\sup_{\nu\in\mathcal{D}_M} R_T^\nu \ge \sup_{\varepsilon\in[0,1]} \sum_{t=13}^T \mathbb{E}\Big[\mathrm{G}_{\varepsilon,t}(m_\varepsilon) - \mathrm{G}_{\varepsilon,t}\big(\alpha_t(V_{\varepsilon,1}, \ldots, V_{\varepsilon,2(t-1)})\big)\Big]$$

$$\ge \sup_{\varepsilon\in[0,1]} \sum_{t=13}^T \mathbb{E}\Big[\mathrm{G}_{\varepsilon,t}(m_\varepsilon) - \mathrm{G}_{\varepsilon,t}\big(\widetilde{\alpha}_t(V_{\varepsilon,1}, \ldots, V_{\varepsilon,2(t-1)})\big)\Big] \stackrel{\spadesuit}{=} \sup_{\varepsilon\in[0,1]} \sum_{t=13}^T \mathbb{E}\left[2\left(\frac{1}{2} - F_\varepsilon\big(\widetilde{\alpha}_t(V_{\varepsilon,1}, \ldots, V_{\varepsilon,2(t-1)})\big)\right)^2\right]$$

$$\stackrel{\circ}{\ge} \sum_{t=13}^T \mathbb{E}\left[2\left(\frac{1}{2} - F_E\big(\widetilde{\alpha}_t(V_{E,1}, \ldots, V_{E,2(t-1)})\big)\right)^2\right] \stackrel{\clubsuit}{=} 2\sum_{t=13}^T \mathbb{E}\left[\left(\frac{5-2E}{8} - \widetilde{\alpha}_t(V_{E,1}, \ldots, V_{E,2(t-1)})\right)^2\right]$$

$$\stackrel{\heartsuit}{\ge} 2\sum_{t=13}^T \mathbb{E}\left[\left(\frac{5-2E}{8} - \mathbb{E}\left[\frac{5-2E}{8} \mid B_{E,1}, \ldots, B_{E,2(t-1)}, D_1, \ldots, D_{2(t-1)}, U_1, \ldots, U_{2(t-1)}\right]\right)^2\right]$$

$$\stackrel{\spadesuit}{=} 2\sum_{t=13}^T \mathbb{E}\left[\left(\frac{5-2E}{8} - \mathbb{E}\left[\frac{5-2E}{8} \mid B_{E,1}, \ldots, B_{E,2(t-1)}\right]\right)^2\right] = \frac{1}{8}\sum_{t=13}^T \mathbb{E}\left[\big(E - \mathbb{E}\left[E \mid B_{E,1}, \ldots, B_{E,2(t-1)}\right]\big)^2\right]$$

$$\stackrel{*}{=} \frac{1}{8}\sum_{t=13}^T \mathbb{E}\left[\left(E - \frac{\sum_{s=1}^{2(t-1)} B_{E,s} + 1}{2t}\right)^2\right] = \frac{1}{8}\sum_{t=13}^T \int_0^1 \mathbb{E}\left[\left(\varepsilon - \frac{\sum_{s=1}^{2(t-1)} B_{\varepsilon,s} + 1}{2t}\right)^2\right]\mathrm{d}\varepsilon$$

$$= \frac{1}{8}\sum_{t=13}^T \int_0^1 \mathbb{E}\left[\left(\varepsilon - \frac{\sum_{s=1}^{2(t-1)} B_{\varepsilon,s}}{2(t-1)} + \frac{\sum_{s=1}^{2(t-1)} B_{\varepsilon,s}}{2(t-1)} - \frac{\sum_{s=1}^{2(t-1)} B_{\varepsilon,s}}{2t} - \frac{1}{2t}\right)^2\right]\mathrm{d}\varepsilon$$

$$= \frac{1}{8}\sum_{t=13}^T \int_0^1 \mathbb{E}\left[\left(\varepsilon - \frac{\sum_{s=1}^{2(t-1)} B_{\varepsilon,s}}{2(t-1)} + \frac{1}{2t(t-1)}\sum_{s=1}^{2(t-1)} B_{\varepsilon,s} - \frac{1}{2t}\right)^2\right]\mathrm{d}\varepsilon$$

$$\geq \frac{1}{8}\sum_{t=13}^{T}\int_0^1 \mathbb{E}\left[\left(\varepsilon - \frac{\sum_{s=1}^{2(t-1)} B_{\varepsilon,s}}{2(t-1)}\right)^2 - 2\left|\varepsilon - \frac{\sum_{s=1}^{2(t-1)} B_{\varepsilon,s}}{2(t-1)}\right|\left|\frac{1}{2t(t-1)}\sum_{s=1}^{2(t-1)} B_{\varepsilon,s} - \frac{1}{2t}\right|\right]\mathrm{d}\varepsilon$$

$$\geq \frac{1}{8}\sum_{t=13}^{T}\int_0^1 \mathbb{E}\left[\left(\varepsilon - \frac{\sum_{s=1}^{2(t-1)} B_{\varepsilon,s}}{2(t-1)}\right)^2 - \frac{1}{t}\left|\varepsilon - \frac{\sum_{s=1}^{2(t-1)} B_{\varepsilon,s}}{2(t-1)}\right|\right]\mathrm{d}\varepsilon$$

$$\overset{\star}{\geq} \frac{1}{8}\sum_{t=13}^{T}\int_0^1\left(\frac{\mathrm{Var}(B_{\varepsilon,1})}{2(t-1)} - \frac{1}{t}\sqrt{\frac{\mathrm{Var}(B_{\varepsilon,1})}{2(t-1)}}\right)\mathrm{d}\varepsilon = \frac{1}{8}\sum_{t=13}^{T}\int_0^1\left(\frac{\varepsilon(1-\varepsilon)}{2(t-1)} - \frac{1}{t}\sqrt{\frac{\varepsilon(1-\varepsilon)}{2(t-1)}}\right)\mathrm{d}\varepsilon$$

$$= \frac{1}{8}\sum_{t=13}^{T}\left(\frac{1}{12(t-1)} - \frac{\pi}{8t\sqrt{2(t-1)}}\right) \geq \frac{1}{8}\left(\frac{1}{12} - \frac{\pi}{16\sqrt{6}}\right)\sum_{t=12}^{T-1}\frac{1}{t} \geq \frac{1}{8}\left(\frac{1}{12} - \frac{\pi}{16\sqrt{6}}\right)\int_{12}^{T}\frac{1}{s}\,\mathrm{d}s$$

$$= \frac{1}{8}\left(\frac{1}{12} - \frac{\pi}{16\sqrt{6}}\right)\ln\left(\frac{T}{12}\right) \geq \frac{1}{16}\left(\frac{1}{12} - \frac{\pi}{16\sqrt{6}}\right)\ln(T)\,.$$

where "♠" follows from Lemma 1; "∘" follows from the fact that $E$ and $V_{\varepsilon,1}, \ldots, V_{\varepsilon,2(t-1)}$ are independent of each other; "♣" follows from the fact that $\widetilde{\alpha}_t$ takes values in $\left[\frac{1}{8}, \frac{7}{8}\right]$ and the explicit formula of $F_\varepsilon$ in that interval for any $\varepsilon \in [0,1]$; "♥" follows from the fact that $\widetilde{\alpha}_t(V_{E,1}, \ldots, V_{E,2(t-1)})$ is $\mathcal{F}_t \coloneqq \sigma(B_{E,1}, \ldots, B_{E,2(t-1)}, D_1, \ldots, D_{2(t-1)}, U_1, \ldots, U_{2(t-1)})$-measurable and that, for any $Y$ the minimizer in $L^2(\mathcal{F}_t)$ of the functional $X \mapsto \mathbb{E}\left[(Y - X)^2\right]$ is $X = \mathbb{E}[Y \mid \mathcal{F}_t]$; "♦" follows from the fact that $E$ and $(D_1, \ldots, D_{2(t-1)}, U_1, \ldots, U_{2t-1})$ are independent of each other; "⋆" follows from the fact $E \mid B_{E,1}, \ldots, B_{E,2(t-1)}$ has a beta distribution; and "★" follows from the fact that $B_{\varepsilon,1}, B_{\varepsilon,2}, \ldots$ is an i.i.d. Bernoulli process of parameter $\varepsilon$, together with Jensen's inequality.

## D   PROOF OF THEOREM 3

Without loss of generality, we can (and do!) assume that $T \geq 2$, and so $\log_2(MT) \geq 1$. First, let $\tau_T$ be the final value of $\tau$ if the algorithm ends at time $T$ without a break, or define it as $\tau - 1$ if it ends with a break. For each $\tau \in [\tau_T]$, we define the epoch $\tau$ as the collection of rounds from $t_{\tau-1} + 1$ to $t_\tau$. Notice that, for each $\tau \in [\tau_T]$, we have that $s_\tau$ is the number of bits collected during the epoch $\tau$. Let $Q_1^\star \coloneqq 1/2$, and define by induction $Q_{\tau+1}^\star$ as $Q_\tau^\star + \frac{1}{2^{\tau+1}}$ if $F(Q_\tau^\star) < 1/2$, as $Q_\tau^\star - \frac{1}{2^{\tau+1}}$ if $F(Q_\tau^\star) > 1/2$, or as $Q_\tau^\star$ if $F(Q_\tau^\star) = 1/2$. If there is $\tau \in \mathbb{N}$ such that $F(Q_\tau^\star) = 1/2$, let $m \coloneqq Q_\tau^\star$. Otherwise, let $m \in [0,1]$ be such that $F(m) = 1/2$ (its existence has already been pointed out after Lemma 1). Crucially, notice that for each $\tau \in \mathbb{N}$, we have that $|m - Q_\tau^\star| \leq 2^{-\tau}$.

Let $(V_{x,k})_{x\in[0,1],k\in\mathbb{N}}$ be an independent family of random variables with common distribution given by $\nu$, and for each $x \in [0,1]$ and $t \in \mathbb{N}$, define $N_t(x) \coloneqq 2 \cdot \sum_{k=1}^{t-1}\mathbb{I}\{P_k = x\}$. Notice that without loss of generality, we can assume that for each $t \in \mathbb{N}$ it holds that $V_{2t-1} \coloneqq V_{P_t, N_t(P_t)+1}$ and $V_{2t} \coloneqq V_{P_t, N_t(P_t)+2}$. Define the "good" event

$$\mathcal{E} \coloneqq \bigcap_{i=1}^{T}\bigcap_{\substack{j=1 \\ j \text{ even}}}^{T}\left\{\left|\frac{1}{j}\sum_{k=1}^{j}\mathbb{I}\{V_{Q_i^\star,k} \leq Q_i^\star\} - F(Q_i^\star)\right| < \sqrt{\frac{\ln(2/\delta)}{2j}}\right\},$$

and notice that by De Morgan's laws, a union bound, and Hoeffding's inequality, we can upper bound the probability of the "bad" event $\mathcal{E}^c$ by $\mathbb{P}[\mathcal{E}^c] \leq \delta T^2$. Notice that for each $i, j \in [T]$ with $F(Q_i^\star) \neq \frac{1}{2}$ and $j$ even satisfying $j \geq \frac{2\ln(2/\delta)}{(\frac{1}{2} - F(Q_i^\star))^2}$, then, whenever we are in the good event $\mathcal{E}$, we have that

$$\frac{1}{j}\sum_{k=1}^{j}\mathbb{I}\{V_{Q_i^\star,k} \leq Q_i^\star\} + \sqrt{\frac{\ln(2/\delta)}{2j}} < F(Q_i^\star) + \sqrt{\frac{2\ln(2/\delta)}{j}} \leq \frac{1}{2}\,,$$

whenever $F(Q_i^\star) < 1/2$, while

$$\frac{1}{j}\sum_{k=1}^{j}\mathbb{I}\{V_{Q_i^\star,k} \leq Q_i^\star\} - \sqrt{\frac{\ln(2/\delta)}{2j}} > F(Q_i^\star) - \sqrt{\frac{2\ln(2/\delta)}{j}} \geq \frac{1}{2}\,.$$

whenever $F(Q_i^\star) > 1/2$. Instead, if $i, j \in [T]$ with $F(Q_i^\star) = \frac{1}{2}$ and $j$ is even, we have that

$$\frac{1}{j}\sum_{k=1}^{j}\mathbb{I}\{V_{Q_i^\star,k} \leq Q_i^\star\} + \sqrt{\frac{\ln(2/\delta)}{2j}} \geq F(Q_i^\star) = \frac{1}{2}$$

and analogously

$$\frac{1}{j}\sum_{k=1}^{j}\mathbb{I}\{V_{Q_i^\star,k} \le Q_i^\star\} - \sqrt{\frac{\ln(2/\delta)}{2j}} \le F(Q_i^\star) = \frac{1}{2}.$$

In particular, if we are in the good event $\mathcal{E}$, these inequalities imply on the one hand that $Q_1 = Q_1^\star, \ldots, Q_{\tau_T} = Q_{\tau_T}^\star$ and, if $\tau \in [\tau_T]$ is such that $F(Q_\tau^\star) = 1/2$, then $\tau = \tau_T$. On the other hand, if $\tau \in [\tau_T]$ is such that $F(Q_\tau^\star) \ne 1/2$ and we are in the good event $\mathcal{E}$, they imply that the number of bits $s_\tau$ collected during the epoch $\tau$ cannot be greater than $\frac{2\ln(2/\delta)}{\left(\frac{1}{2}-F(Q_\tau^\star)\right)^2}$, because the condition that ends the epoch $\tau$ with a break is met by the time that we have collected $\frac{2\ln(2/\delta)}{\left(\frac{1}{2}-F(Q_\tau^\star)\right)^2}$ bits in that epoch.

Define $\tau_T^\# \coloneqq \lceil \log_2(MT) \rceil$, define $\tau_T^\flat$ as the smallest $\tau \in \mathbb{N}$ such that $F(Q_\tau^\star) = 1/2$ if it exists, and $+\infty$ otherwise, and define $\tau_T^\star \coloneqq \min(\tau_T^\#, \tau_T^\flat, \tau_T)$. In what follows, when we are in the event $\tau_T^\# > \max(\tau_T^\flat, \tau_T)$, we use the convention that any summation of the form $\sum_{\tau=\tau_T^\star+1}^{\tau_T}$ is zero by definition. For each $t \in [T]$, define $\mathcal{H}_t \coloneqq \sigma(V_1, \ldots, V_{2t-2})$ as the $\sigma$-algebra generated by the history observed before time $t$. We can control the regret in the following way

$$R_T = \sum_{t=1}^{T}\mathbb{E}\big[G_t(m) - G_t(P_t)\big] = \sum_{t=1}^{T}\mathbb{E}\Big[\mathbb{E}\big[G_t(m) - G_t(P_t) \mid \mathcal{H}_t\big]\Big]$$

$$\overset{\spadesuit}{=} \sum_{t=1}^{T}\mathbb{E}\bigg[\Big[\mathbb{E}\big[G_t(m) - G_t(p)\big]\Big]_{p=P_t}\bigg] \overset{\clubsuit}{=} 2\cdot\sum_{t=1}^{T}\mathbb{E}\bigg[\Big(\frac{1}{2} - F(P_t)\Big)^2\bigg]$$

$$\le 2\cdot\sum_{t=1}^{T}\mathbb{E}\bigg[\Big(\frac{1}{2} - F(P_t)\Big)^2\mathbb{I}_\mathcal{E}\bigg] + \frac{T}{2}\cdot\mathbb{P}[\mathcal{E}^c]$$

$$= \mathbb{E}\bigg[\sum_{\tau=1}^{\tau_T^\star-1} s_\tau\cdot\Big(\frac{1}{2} - F(Q_\tau^\star)\Big)^2\mathbb{I}_\mathcal{E}\bigg] + \mathbb{E}\bigg[\sum_{\tau=\tau_T^\star}^{\tau_T} s_\tau\cdot\Big(\frac{1}{2} - F(Q_\tau^\star)\Big)^2\mathbb{I}_\mathcal{E}\bigg] + \frac{T}{2}\cdot\mathbb{P}[\mathcal{E}^c]$$

$$\overset{\heartsuit}{\le} \mathbb{E}\bigg[\sum_{\tau=1}^{\tau_T^\star-1} \frac{2\ln(2/\delta)}{\left(\frac{1}{2}-F(Q_\tau^\star)\right)^2}\cdot\Big(\frac{1}{2} - F(Q_\tau^\star)\Big)^2\mathbb{I}_\mathcal{E}\bigg] + \mathbb{E}\bigg[\sum_{\tau=\tau_T^\star}^{\tau_T} s_\tau\cdot M^2\cdot|m - Q_\tau^\star|^2\bigg] + \frac{T}{2}\cdot\mathbb{P}[\mathcal{E}^c]$$

$$\le (\tau_T^\# - 1)\cdot 2\cdot\ln(2/\delta) + T\cdot M^2\cdot 2^{-2\tau_T^\#} + \delta\cdot\frac{T^3}{2} \le 2 + 6\log_2(MT)\ln(T),$$

where in $\spadesuit$ we used the Freezing Lemma (see, e.g.,(Cesari & Colomboni, 2021, Lemma 8)), in $\clubsuit$ we used Lemma 1, and in $\heartsuit$ we used that fact that $F(m) = 1/2$ and $F$ is $M$-Lipschitz.

## E   PROOF OF THEOREM 4

We already know that algorithms that have access to full-feedback have to suffer worst-case regret of at least $c_1 \ln T$ if $T \ge c_2$, where $c_1$ and $c_2$ are the constants in the statement of Theorem 2. In particular, the same statement holds *a fortiori* for any 2-bit feedback algorithm, given that any 2-bit feedback algorithm can be trivially converted into an algorithm operating with full-feedback. It follows that it is enough to prove that there exist two universal constants $\widetilde{c}_1$ and $\widetilde{c}_2$ such that the worst-case regret of any 2-bit feedback algorithm is at least $\widetilde{c}_1 \ln M$ whenever $T \ge \widetilde{c}_2 \log_2(M)$. In fact, in this case, we can set $\bar{c}_1 \coloneqq \frac{1}{2}\min(c_1, \widetilde{c}_1)$ and $\bar{c}_2 \coloneqq \max(c_2, \widetilde{c}_2)$ to obtain that the worst-case regret of any 2-bit feedback algorithm is at least $2\bar{c}_1\max(\ln T, \ln M) \ge \bar{c}_1\ln(MT)$ whenever $T \ge \bar{c}_2\log_2 M$.

We now prove the existence of $\widetilde{c}_1$ and $\widetilde{c}_2$. Let $n \in \mathbb{N}$ be the greatest integer such that $2^n \le M$ and consider the elements $\nu_k \in \mathcal{D}_M$ whose density is $2^n\cdot\mathbb{I}_{\left(\frac{k-1}{2^n}, \frac{k}{2^n}\right)}$ for some $k \in [2^n]$, and notice that the corresponding cdfs are $M$-Lipschitz.

Consider the following surrogate game. The adversary secretly chooses $k^\star \in [2^n]$. The player action space is $[2^n]$. The surrogate game ends the first time $t \in \mathbb{N}$ when the player plays $I_t = k^\star$. Before that, if the player plays $I_t \ne k^\star$, the player suffers a loss $1/2$ and receives $\mathbb{I}\{I_t \le k^\star\}$ as feedback. Now, note that we can convert any algorithm $\alpha$ for the 2-bit feedback problem into an algorithm $\widetilde{\alpha}$ for the surrogate game in the following way. For each $k \in [2^n - 1]$, define $J_k \coloneqq [(k-1)2^{-n}, k2^{-n})$ and $J_{2^n} \coloneqq [(2^n - 1)2^{-n}, 1]$. Whenever the algorithm $\alpha$ plays $P_t \in J_k$, the algorithm $\widetilde{\alpha}$ plays $I_t \coloneqq k$

and passes $\left(\mathbb{I}\{I_t \leq k^\star\}, \mathbb{I}\{I_t \leq k^\star\}\right)$ to $\alpha$, where $k^\star$ is the underlying instance of the surrogate game. Now, notice that we can map every instance $k^\star \in [2^n]$ for the surrogate game into the instance $\nu_{k^\star} \in \mathcal{D}_M$ of the original problem and that the regret of the algorithm $\alpha$ on the instance $\nu_{k^\star}$ is greater than or equal to than the regret of the algorithm $\widetilde{\alpha}$ on the instance $k^\star$. It follows that a worst-case regret lower bound for the surrogate game is also a worst-case regret lower bound for the original problem.

Fix an algorithm $\alpha$ for the surrogate game. Given that the surrogate game is deterministic, without any loss of generality we can assume that $\alpha$ is deterministic. We say that $S \subset [2^n]$ is a discrete segment if $S$ is of the form $\{k \in [2^n] \mid a \leq k \leq b\}$ for some $a, b \in [2^n]$ with $a \leq b$. We can prove the following property by induction on $t = 0, 1, \ldots, n - 1$: there is a discrete segment $J_t$ with at least $2^{n-t} - 1$ elements such that, for each $k, k' \in S_t$, the algorithm has not won the game by the time $t$ and receives the same feedback (and hence selects the same actions) if the underlying instance is $k$ or $k'$. For $t = 0$ the property is true by setting $S_0 := [2^n]$. Assume that the property is true for some $t \in \{0, 1, \ldots, n - 2\}$. Assume that $a, b \in [2^n]$ with $a \leq b$ are such that $S_t = \{k \in [2^n] \mid a \leq k \leq b\}$, where $S_t$ is a segment that enjoys the property. Now, if the algorithm plays $I_{t+1} \notin S_t$ we can set $S_{t+1} := S_t$, and we see that the required properties hold trivially. Instead, if $I_{t+1} \in S_t$ we set $S_{t+1} := \{k \in [2^n] \mid I_t + 1 \leq k \leq b\}$ if $I_{t+1} < \frac{a+b}{2}$ and we set $S_{t+1} := \{k \in [2^n] \mid a \leq k \leq I_t - 1\}$ if $I_{t+1} \geq \frac{a+b}{2}$. Notice that given that $S_t$ has at least $2^{n-t} - 1$ points, we have that $S_{t+1}$ contains at least $\frac{2^{n-t}-2}{2} = 2^{n-(t+1)} - 1$ points and, for each $k \in S_{t+1}$, the game does not end by the time $t + 1$. Hence the induction step is proved. It follows that $S_{n-1}$ is non-empty and, if we pick $k^\star \in S_{n-1}$, the game goes on at least up to time $n - 1$ whenever the time horizon $T$ is at least $n - 1$. Hence, if $T \geq \log_2(M)$ (which implies in particular that $T \geq n - 1$), the worst-case regret of the algorithm $\alpha$ is at least $\frac{n-1}{2} = \frac{n+1}{2} - 1 \geq \frac{\log_2(M)}{2} - 1 \geq \frac{\log_2(M)}{4} = \frac{1}{4\ln(2)} \ln M$, where in the last inequality we used $M \geq 16$. Hence, we can pick $\widetilde{c}_1 := \frac{1}{4\ln 2}$ and $\widetilde{c}_2 := 1$, concluding the proof.

## F  PROOF OF THEOREM 6

For any $t \in \mathbb{N}$, tedious but straightforward computations show that

$$\mathbb{P}\left[\sup_{p \in [0,1]} \left|\Psi(p) - \hat{\Psi}_t(p)\right| \geq \varepsilon\right] \leq \mathbb{P}\left[\sup_{p \in \mathbb{R}} \left|\frac{1}{2t}\sum_{s=1}^{2t}\mathbb{I}\{V_s \leq p\} - F(p)\right| \geq \frac{\varepsilon}{4}\right] \leq 2\exp\left(-\frac{1}{4}\varepsilon^2 t\right),$$

where the last inequality follows from the DKW inequality (Massart, 1990). Let $p^\star \in \arg\max_{p \in [0,1]} \Psi(p)$ (which does exist due to the upper-semicontinuity of $\Psi$). Then, for any $t \in \mathbb{N}$, we have that

$$\mathbb{E}\left[\Psi(p^\star) - \Psi(P_{t+1})\right] = \mathbb{E}\left[\Psi(p^\star) - \hat{\Psi}_t(p^\star)\right] + \underbrace{\mathbb{E}\left[\hat{\Psi}_t(p^\star) - \hat{\Psi}_t(P_{t+1})\right]}_{\leq 0} + \mathbb{E}\left[\hat{\Psi}_t(P_{t+1}) - \Psi(P_{t+1})\right]$$

$$\leq 2\mathbb{E}\left[\sup_{p \in [0,1]} \left|\Psi(p) - \hat{\Psi}_t(p)\right|\right] = 2\int_0^{+\infty} \mathbb{P}\left[\sup_{p \in [0,1]} \left|\Psi(p) - \hat{\Psi}_t(p)\right| \geq \varepsilon\right] d\varepsilon$$

$$\leq 2\int_0^{+\infty} 2\exp\left(-\frac{1}{4}\varepsilon^2 t\right) d\varepsilon = \frac{4\sqrt{\pi}}{\sqrt{t}}.$$

Hence

$$R_T \leq 1 + \mathbb{E}\left[\sum_{t=2}^{T}\left(\Psi(p^\star) - \Psi(P_t)\right)\right] \leq 1 + 4\sqrt{\pi}\sum_{t=1}^{T-1}\frac{1}{\sqrt{t}} \leq 1 + 8\sqrt{\pi}\cdot\sqrt{T-1}.$$

