# OpenReview forum: "An Online Learning Theory of Trading-Volume Maximization"
_ICLR.cc/2025/Conference — ICLR 2025 Poster_

### Official Review · Reviewer_6KDK · 2024-10-25

**Soundness:** 4
**Presentation:** 4
**Contribution:** 3
**Rating:** 8
**Confidence:** 4

**Summary:**

This paper focuses on the bilateral trade problem in an online learning setting. Specifically, the authors consider a setting where, at each time step, a buyer and a seller arrive, and the algorithm, acting as the broker, needs to decide on a price for the trade. The trade is successful if the buyer's value is higher than the price, and the seller's value is lower than the price. Unlike prior works on bilateral trade that focused on optimizing **gain from trade**, the authors study the problem under a new measure of performance: **total number of trades**. The buyer's and the seller's values are drawn from an unknown distribution i.i.d. at each time step. The authors study two models of the algorithm's observation: after each time step, it observes either both players' values or their willingness to sell or buy at each time step. Under both models, and under different assumptions on the distribution of the values, the authors provide algorithms with guarantees on their regret and lower bounds on the regret of any algorithm. In most cases, the regret guarantees and lower bounds match each other asymptotically.

**Strengths:**

1. The models considered in this paper are natural, and the theoretical assumptions are mostly reasonable.
2. The theoretical results are mostly tight and elegantly shown in Table 1.
3. The paper is quite well-written and easy to follow.

**Weaknesses:**

I have no major concerns about this paper and am happy to recommend it for acceptance. However, I have a few minor comments that the authors may consider addressing. Please also refer to the question section for some questions that I have. Feel free to ignore them since I am already positive about the paper.

1. **Impact.** The paper presents a number of theoretical results, and it is commendable that they are almost tight. However, these results are mostly derived from standard techniques. This makes it unclear how this work will inspire future research from a technical perspective.
2. **Related work.** The discussion on related work about the original (non-online-learning) bilateral trade problem is a bit limited from my perspective. Currently, it is mostly stacked citations in one paragraph without sufficient explanation (even Myerson & Satterthwaite, 1983). It would be helpful to have a bit more discussion on the original bilateral trade problem and how it is related to the setting in this paper. Moreover, a few references are missing, such as:
   - The Gains from Trade Under Fixed Price Mechanisms (Applied Economics Research Bulletin 2008)
   - Improved Approximation to First-Best Gains-from-Trade (WINE 2022)
   - Non-excludable Bilateral Trade between Groups (AAAI 2024)

**Questions:**

1. Both the buyer and the seller share the same value distribution. Can you further discuss the motivation behind this assumption? What would happen if they have different value distributions?
2. Does any prior work on bilateral trade consider the total number of trades as the performance measure?
3. How do you think the results in this paper will impact the community? Is it more about the techniques or the problem setting?

---

> ### Author Response · Authors · 2024-11-20
>
> **Related work**
>
> We thank the reviewer for pointing out the additional related work. We will add them to the revised version. We are also open to expand the beginning of the related work section as suggested. We agree that it is currently a tad too compressed.
>
> **Q1. Same distribution**
>
> The key difference from the classical bilateral trade problem is that here, there are no designated "buyers" and "sellers" populations, but a single "traders" population exists. We elaborate on this in the general answer, as other reviewers asked similar questions. Please, check it out and if you have any further questions, do not hesitate to ask!
>
> **Q2. Prior work considering the total number of trades as the performance measure**
>
> To the best of our knowledge, we are the first to analyze this performance measure.
> Anecdotally, from the moment bilateral trade problems started to get traction in the online learning community, this performance measure has been brought up many times in conferences/workshops/seminars in which bilateral trade works have been discussed.
> This is what originally motivated us to attempt investigating it.
>
> **Q3. Impact**
>
> In terms of impact, the aspect we feel most strongly about is that trading-volume maximization drives more fairness in trading because it does not sacrifice traders who live off small margins.
> Another positive aspect where trading volume differs from gain from trade is that it is an observable metric, which can be quantified without the traders revealing their valuations and, as such, can be more transparently tracked.
> As an additional positive side-effect, maximizing trading volume leads to consistent regret minimization speed-ups (i.e., increased efficiency in maximizing the objective) that can't be otherwise obtained if the metric is the gain from trade. (E.g., in the $2$-bit feedback setting, it is possible to minimize the regret at a logarithmic rate, exponentially faster than the corresponding $\sqrt{T}$ when the objective is the gain from trade.)
>
> Another aspect of our work that we hope will inspire future research is the care of having lower bounds that match in all relevant parameters. Most notably, optimality with respect to the Lipschitz constant of the cdfs is seldom studied, as there are no standard techniques that allow to derive it and require ad hoc analyses depending on the variants of the problem at hand.

---

> > ### Comment · Reviewer_6KDK · 2024-11-20
> >
> > Thank you for your detailed reply to my review. It helps to clarify the questions I had during the review process, and I am glad you would improve the beginning of the related work section. Moreover, the elaboration of the paper's potential impact is particularly compelling. I am convinced by the argument that trading-volume maximization is a fairer metric, and the work's impact is therefore clear, given that it is the first one to analyze the metric. I have increased my score to 8 in light of your response.

---

> > > ### Author Response · Authors · 2024-11-20
> > >
> > > We really appreciate you kind words! Thanks!

---

### Official Review · Reviewer_TXbV · 2024-11-01

**Soundness:** 4
**Presentation:** 3
**Contribution:** 2
**Rating:** 6
**Confidence:** 3

**Summary:**

The authors study the online learning to maximize trading volume. Two traders come with private valuation in pairs, and the brokerage posts a price. If the price lies in between the valuation of the pair of traders a trade takes place. They study full information and 2-bit feedback structure. In full information, the valuation of two traders are revealed, while in 2-bit feedback whether the trader's valuation is higher than the posted price are revealed. They study the regret guarantees in three settings - the cdf of the reward is  (a) Lipschitz, (b) Continuous, and (c) General. In full information they provide tight regret bounds for all three settings. In 2-bit feedback Lipschitz functions admit tight logarithmic regret bounds, while linear regret lower bounds are established for the other two settings.

**Strengths:**

- The characterization of regret in the different setting seems quite complete.
- The study of the online trading-volume maximization is an interesting addition to online learning in bilateral trade.

**Weaknesses:**

- The results though complete seems a bit lacking in terms of technical challenges. Simple median based algorithms suffice.
    - Complementing the general setting with $\alpha$-regret (for some reasonable $\alpha$) would improve the paper.
    -  Coverage of the 1-bit feedback (whether trade went through or not) would improve the paper.
- Incentive compatibility of the proposed technique is not discussed.
- There are some work in bandit learning in online Auctions/double auctions. Is there any connection to that work?

**Questions:**

See weaknesses.

---

> ### Author Response · Authors · 2024-11-20
>
> **Technical challenges**
>
> We appreciate the reviewer’s observation regarding the simplicity of the high-level algorithmic ideas, but we see this as a strength rather than a weakness of our work.
> Presenting simple and intuitive ideas was possible only after *extensive* refinements and it is not necessarily indicative of a lack of technical challenges.
> The process of identifying the right approach was not straightforward, but we understand that presenting the final algorithms and analysis in a clear and accessible manner inevitably conceals the underlying development process, which was far from trivial.
>
> Moreover, even for intuitive algorithmic ideas, *proving* that their guarantees are optimal was challenging.
> We hope the reviewer acknowledges that the proofs of both the upper and lower bound guarantees, which establish the optimality of these algorithms, required a nuanced and technically demanding analysis.
>
>
> **Extensions**
>
> The reviewer also expressed curiosity about potential directions to extend our work.
>
> **$\alpha$-regret**
>
> To begin with, the reviewer inquired about what can be said if the notion of regret is weakened to $\alpha$-regret.
> We note that $\alpha$-regret is a meaningful benchmark only in scenarios where the problem is unlearnable under the standard regret framework.
> In our setting, this occurs in the $2$-bit feedback scenario when the Lipschitz constant of the cumulative distribution function can vary with the time horizon (or when the cdf is not Lipschitz), as we proved that the problem is unlearnable in this case (Theorem 7).
> To illustrate why meaningful $\alpha$-regret guarantees are unattainable for any $\alpha$ in this case, we provide the following intuitive argument.
> Consider a highly concentrated valuation density for the traders, such as the normalized indicator function $f_{r,\varepsilon}$ of an interval centered at $r$ with length $\varepsilon$.
> For any algorithm, we can find a suitable $r$ such that the algorithm requires at least $\Omega(\log(1/\varepsilon))$ queries to locate the region where the density mass is concentrated.
> During this period, no trades occur, resulting in zero gain for the algorithm in these rounds.
> In contrast, a trader with knowledge of the underlying density $f_{r,\varepsilon}$ could post $r$ immediately, achieving a trading probability of $1/4$.
> Consequently, for any time horizon $T$ and any algorithm, there exist some $r$ and sufficiently small $\varepsilon$ such that no multiplicative constant $\alpha$ can bridge the gap between the learner’s earnings (zero) and those of an oracle with knowledge of the distribution ($T/4$).
>
> **$1$-bit feedback**
>
> The reviewer also inquired about the case where, after each interaction, the learner has access only to a single bit of feedback indicating whether a trade occurred at the proposed price.
>
> We did not consider this type of feedback in our work because it is arguably less relevant than the $2$-bit feedback (corresponding to posted-price mechanisms) and full feedback (corresponding to direct revelation mechanisms), which are by far two most studied settings in the economics literature.
> Additionally, it is difficult to envision real-life scenarios where the broker has access only to a binary indicator of whether a trade occurred, rather than the two individual responses from the traders with whom the interaction takes place.
> Nevertheless, we acknowledge that this is an interesting mathematical question.
> First, we note that this $1$-bit feedback is inherently less informative than the $2$-bit feedback, which we already proved to be unlearnable when we drop the $M$-Lipschitz cdf assumption.
> Therefore, we discuss this problem under the assumption that the cdf of the valuations is $M$-Lipschitz.
> Second, we note that this feedback is still sufficient to reconstruct bandit feedback.
> Hence, a natural approach in this case could involve discretization: we could build a uniform grid with $K$ price points and use an optimal bandit algorithm, treating these prices as arms.
> This strategy incurs a regret of at most $\sqrt{KT}$ from the bandit algorithm, along with additional regret from the discretization error.
> Leveraging our Lemma 1, the fact that $F(m) = 1/2$, and the assumption that $F$ is $M$-Lipschitz, we can bound the regret due to discretization by $\frac{M^2}{K^2}T$.
> Balancing these terms by choosing $K = M^{4/5}T^{1/5}$ leads to sublinear regret guarantees of order $O(M^{2/5}T^{3/5})$.
>
> We believe that determining whether this algorithm is optimal (for example, by establishing a suitable matching lower bound) is an intriguing question.
> Notably, as demonstrated in other studies on bilateral trade with a continuum of arms, exotic rates like this could, in principle, emerge, deviating from the standard $\sqrt{T}$, $T^{2/3}$, and $T$ trichotomy observed in finite partial monitoring.
> We leave this exploration as an interesting open direction for future research.

---

> > ### Author Response · Authors · 2024-11-20
> >
> > **Incentive compatibility**
> >
> > The reviewer also inquired about the incentive compatibility of our proposed mechanism.
> > We emphasize that, consistent with the broader body of bilateral trade literature studied from an online learning perspective, our mechanism is truthful for traders who make "take-it-or-leave-it" decisions and subsequently leave the broker permanently. In such scenarios, traders have no incentive to misreport their valuations, as doing so would yield no benefit and could instead result in forfeiting the opportunity to secure a non-negative utility.
> > This assumption aligns naturally with the i.i.d.\ framework, which can be interpreted as representing traders drawn from a large and stable market. In such markets, re-entry effects are negligible, and the distribution of traders' valuations remains essentially unchanged over time.
> > At the same time, addressing the game-theoretic challenges posed by strategic traders who might return over time is undoubtedly an interesting and important open problem.
> > However, as with the rest of the online learning literature on bilateral trade, we believe this topic lies beyond the scope of the present study and is better suited for future research.
> >
> > **Connections with auctions**
> >
> > Finally, the reviewer asked about possible connections with bandit learning in online auctions or double auctions.
> > As we highlighted in the techniques and challenges section, standard bandit methods—such as discretization and exponential weights—can indeed be applied to our problem. However, these approaches typically lead to suboptimal guarantees and often require additional assumptions (with the potential exception of the 1-bit feedback case discussed above).
> > If the reviewer has specific works in mind, we would be happy to address them directly.

---

> > ### Comment · Reviewer_TXbV · 2024-11-21
> > **Response to rebuttal**
> >
> > I thank the authors for the response. I encourage the the authors to include the discussions around fairness and incentive compatibility. Further, adding the discussions on the $1$-bit feedback and the $\alpha$-regret will improve the paper. I will maintain my score.

---

> > > ### Author Response · Authors · 2024-11-21
> > >
> > > Will do, and thank you again for reviewing our submission!

---

### Official Review · Reviewer_RtL8 · 2024-11-04

**Soundness:** 4
**Presentation:** 3
**Contribution:** 3
**Rating:** 8
**Confidence:** 4

**Summary:**

The paper studies trade volume maximization in brokerages through an online learning perspective. It considers bilateral trades where in each round two traders participate. The brokerage sets a price $p_t$, and the traders independently sample a private valuation from a fixed distribution. The trade happens if the brokerage price is between the two traders' sampled valuations, that is one trader values it below the price and is willing to sell it at this price, and one trader values it more than the price and is willing to buy at the price.

Previous works consider regret minimization aiming to maximize total profits, but this paper introduces a new objective where the goal of the brokerage is to maximize the number of trades.

The authors motivate the objective by showing an example where profit maximization can lead to low trade volumes.

They consider full feedback setting and partial feedback setting.

For the full feedback setting, they show that having continuous CDF is enough to get $O(\ln T)$ regret upper bound. In contrast, for the profit maximization setting, prior work has shown that Lipschitzness of the CDF is required to achieve even $\tilde{O}(T)$ upper bounds. They also show a matching lower bound. For the non-continuous case, they show a $O(\sqrt{T})$ lower bound and give an algorithm that achieves a matching upper bound.

For the partial feedback, the brokerage only learns if the valuations were lower or higher than the price. Under $M$ Lipschitzness of the CDF, the authors give an algorithm with $O(\ln(MT) \ln T)$ upper bound and show an $ \Omega( \ln(MT))$ lower bound. They also show that Lipschitzness is required to obtain sublinear regret in the partial feedback case.

**Strengths:**

1. The profit maximization objective in online trades is well-studied in recent literature, the new objective introduced in this paper considers trade volume maximization and is reasonably motivated.

2. The insight that the i.i.d. assumption in trade volume maximization simplifies the setting into a median estimation problem is novel resulting in tighter upper/lower bounds than the profit maximization objective.

3. The paper provides tight upper and lower bounds in the full feedback setting, and strong results in the partial feedback setting with thorough analysis.

4. The paper does a good job of explaining the related work in this domain.

**Weaknesses:**

1. Even though the authors motivate the trading volume objective, I would also benefit from some discussion on the potential drawbacks of aiming for trading volume maximization. Even in the motivating example in Section 3, when the price is optimized for trading volume maximization, even though it leads to more trades, the total profits the traders achieve are almost halved (from approx $\frac{\epsilon(1-\epsilon)}{2}$ to approx $\frac{\epsilon(1-\epsilon)}{4} + O(\epsilon^2)$) which is linear regret for total profit.

2. Most insights and algorithms are based on the fact that trade volume is maximized at the median in the i.i.d. setting where buyers and sellers have the same valuation distribution. Thus, the techniques may not translate to the generally studied i.i.d setting where the buyer and seller have different and possibly correlated distributions. The authors mention it as a possible future work, but maybe they can motivate the i.i.d setting more.

**Questions:**

1. Can you please talk about the potential drawbacks of trade volume maximization?

2. Can you please discuss how the 2 distribution i.i.d. cases studied in the literature relate to the i.i.d. case with one distribution regarding problem difficulty?

---

> ### Author Response · Authors · 2024-11-20
>
> Dear reviewer,
>
> Thank you for your kind words. Please, see the two general answers that address your comments. If you have any further questions, do not hesitate to ask!

---

> > ### Comment · Reviewer_RtL8 · 2024-11-21
> >
> > Thanks for addressing my comments and adding the discussion.

---

> > > ### Author Response · Authors · 2024-11-21
> > >
> > > You are welcome, and thank _you_ again for your insightful comments!

---

### Official Review · Reviewer_BE5B · 2024-11-04

**Soundness:** 3
**Presentation:** 3
**Contribution:** 2
**Rating:** 6
**Confidence:** 3

**Summary:**

The paper studies a online learning problem for the brokerage who aims to maximize the total trading volume by posting a single price to two sequentially arriving traders. In particular, at each round $t$, there are two traders arriving with their private valuations $V_{2t-1}$ and $V_{2t}$. The broker then posts a price $P_t$ for this round, and if this price $P_t$ is between $V_{2t-1} \vee V_{2t}$ and $V_{2t-1} \wedge V_{2t}$, then the trade happens.
Instead of optimizing the total gain from trade discussed in previous works, the goal of the brokerage of this work is to maximizing the total trading volume — i.e., the total expected trading probability.

The paper assumes that the two valuations of both traders are all independently and identically realized from an unknown distribution. The paper considers two feedback structure. One is full feedback where the brokerage can observe both traders’ valuations $V_{2t-1}, V_{2t}$, another is the 2-bit feedback where the brokerage can only observe indicators $\mathbb{I}\{V_{2t-1}\le P_t\}$ and $\mathbb{I}\{V_{2t}\le P_t\}$.

The paper first shows that in the full-feedback case, if the distribution of the traders’ valuations has a continuous cdf, then an $O(\ln T)$-regret algorithm exists and this is tight. They also show that if we dropthe continuous cdf assumption, then the regret degrades to $\Omega(\sqrt{T})$, and they also provide an algorithm for this regret bound.

In the 2-bit feedback case, if the cdf of the traders’ valuations is M -Lipschitz, they show that  $O(\ln(MT)\lnT)$-regret algorithm exists and also provide a near-matching lower bound $O(\ln(MT))$. The problem becomes becomes unlearnable if one drop the Lipschitzness assumption.

**Strengths:**

The paper studies an online brokerage problem with adopting a new metric compared to current online bilateral trade problems. The new metric measures the expected probability of a successful trade which is motivated by a fairness concern: ``trading-volume maximization gives the same dignity to all traders, granting everybody the same opportunity to trade, independently of their buying power’’. I think taking a step to study the fairness issue in bilateral trade has solid motivation. The paper also provides complete results for this problem (though making some simplifying assumptions).

**Weaknesses:**

Although I agree with that studying the fairness issue in bilateral trade is an interesting problem, I found the paper lack of justifications on why the “trading-volume maximization” is indeed achieving certain fairness, as this only grants the same trading probability for each party. How does metric compared to the metric that is related trader’s welfare? I think more discussions here may be helpful to better motivate the current metric.

Meanwhile, I think the current results are bit restricted given that they assume both traders’ valuations are all realized from a same distribution. This assumption seems to be a bit unpractical in real-world. Would current results/algorithms generalize to this setting?

**Questions:**

Please see above comments.

---

> ### Author Response · Authors · 2024-11-20
>
> Dear reviewer,
>
> Please, see the two general answers that address your comments. If you have any further questions, do not hesitate to ask!

---

> ### Comment · Reviewer_BE5B · 2024-11-21
>
> Dear Authors,
>
> Thanks for your feedback and responses for my questions. It might be good to add those discussions related to the fairness metric in your revision as they are helpful to motivate the current fairness metric. (Feel free to ignore this) For example, you can add a new section to discuss the implication of why trading-volume maximization can indeed achieve certain fairness. To me, I think this alone is an interesting question to explore in more detail.
>
> Also I feel it may be helpful to add those examples of the OTC market that can motivate the same valuation distribution.
>
> I have increased my score as I am satisfied with authors' response.

---

> > ### Author Response · Authors · 2024-11-21
> >
> > Dear reviewer,
> >
> > We completely agree that adding this content would improve the paper and we are happy to commit to this revision. We also appreciate you raising your score and thank you for your input during this reviewing process!

---

### Author Response · Authors · 2024-11-20
**General Comment for all reviewers**

We thank all reviewers who commented on the two points below.
In addition to addressing them here, we will add this discussion to the revised version of the submission as it clarifies and, in fact, strengthens the message of this work.

**General Comment \#1: Fairness and drawbacks of trading-volume maximization.**

To elaborate on the fairness and drawbacks of trading-volume maximization, in particular (as requested) in regards to our motivating example in Section 3, we first compute the individual expected utilities of traders in the motivating example, in the two cases in which the learner posts the price $1/2$ maximizing the gain from trade and the price $m = \frac12 - \frac{\epsilon}2 \frac{1-2\epsilon}{1-\epsilon}$ maximizing the trading volume.
A direct verification shows the following:

1. The selling expected utility of traders at $1/2$ is $\epsilon^2 - \epsilon^3 = \Theta(\epsilon^2)$
2. The buying expected utility of traders at $1/2$ is $\epsilon - 2 \epsilon^2 + \epsilon^3 = \Theta(\epsilon)$
3. Thus, the gain from trade at $1/2$ is $\epsilon - 2 \epsilon^2 = \Theta(\epsilon)$.


On the other hand:

1. The selling expected utility of traders at $m$ is $\frac{1}{8}\epsilon + \frac{1}{4} \epsilon^2 = \Theta(\epsilon)$
2. The buying expected utility of traders at $m$ is $\frac{5}{8} \epsilon = \Theta(\epsilon)$
3. Thus, the gain from trade at $m$ is $\frac{3}{4} \epsilon + \frac{1}{4}\epsilon^2 = \Theta(\epsilon)$.

**Fairness**

Since the distribution of the traders' valuations is continuous and supported on the union $[\frac12-\epsilon, \frac12] \cup [1-\epsilon, 1]$ of a low and a high-valuation cluster (with the overwhelming majority of the population concentrated in the low-valuation cluster), then, posting the price $1/2$ that maximizes the gain from trade has the following consequences:

- Traders from the high-valuation cluster (who enter the market with probability $\Theta(\epsilon)$) will trade with probability $\Theta(1)$ and earn utility $\Theta(1)$.

- Traders from the low-valuation cluster (who enter the market with probability $\Theta(1)$) trade with probability $\Theta(\epsilon)$ and earn utility $\Theta(\epsilon)$.

We argue that this is unfair because traders in the low-valuation cluster (which constitutes the near-totality of the population) are effectively made unable to trade with each other and, as a consequence, end up earning negligible $\Theta(\epsilon^2)$ expected utilities.
In this case, the near-totality of the gain from trade comes from the $\Theta(\epsilon)$ expected utility accrued by the high-valuation traders.

In contrast, posting the price $m$ that maximizes the trading volume has the following consequences.

- Traders from the high-valuation cluster (who enter the market with probability $\Theta(\epsilon)$) will trade with probability $\Theta(1)$ and earn utility $\Theta(1)$.

- Traders from the low-valuation cluster (who enter the market with probability $\Theta(1)$) trade with probability $\Theta(1)$ and earn utility $\Theta(\epsilon)$.

In this case, low-valuation traders can trade with each other (as well as with high-valuation traders), and consequently, their expected utilities rise to $\Theta(\epsilon)$.
In this case, both low and high-valuation traders accrue $\Theta(\epsilon)$ expected utilities, maintaining in particular the same order $\Theta(\epsilon)$ (see next paragraph for further discussion) of expected gain from trade.

**Drawbacks**

As a reviewer correctly remarked, there is no free lunch.
The fairness obtained by moving away from directly maximizing the gain from trade comes at the cost that, in general, the gain from trade won't be exactly maximized.
Take the previous instance as an example.
The trading-volume objective effectively manages to drive a balance of the utilities of traders coming from high and low-valuation clusters of the population but at the price of slightly lowering the gain from trade (note that although the order is the same, the gain from trade at $m$ is lower than the optimal one by a multiplicative constant).
If $\epsilon$ is a constant (however small) independent of the time horizon, this will result in a linear regret with respect to the gain from trade.

This should not come as a surprise, though, as it is what one should expect from "fair" objectives: generally speaking, they are not as efficient as "unfair" ones.
We agree with the reviewer that this is a point worth raising in a broader sense.
*Is it worth losing a fraction of the global value to obtain a fairer distribution of a slightly lower total wealth?*
Perhaps. This is a profound philosophical question that reaches far beyond the scope of our work and whose answer probably varies depending on the circumstances.
Still, we believe it is valuable to investigate both options and, at the very least, shine a light on the possible pros and cons of the two, as well as to flesh out the different techniques that are needed to tackle each variant of the problem.

---

> ### Author Response · Authors · 2024-11-20
>
> **General Comment \#2: Same-distribution assumption**
>
> Unlike bilateral trade settings where traders join the market as either buyers or sellers (in which case, it is natural to assume that the populations of buyers and sellers have distinct valuation distributions), in our setting, each trader has no predetermined intent to buy or sell.
> Instead, they simply hold a valuation for the item on sale and are willing to either buy or sell depending on the price proposed by the broker.
>
> Settings with flexible seller/buyer roles have already appeared in the literature and are prevalent in OTC markets.
> Think of the stock market, or markets of gems and minerals, but also (perhaps less obviously) agricultural markets, as described in [1]:
> "*Consider, for example, markets for agricultural goods [...]. Participants in agricultural markets [...] may switch between seller and buyer roles depending on individual outcomes, consumption needs, and trading opportunities. [...] A considerable literature in development economics studies agricultural households navigating such decisions (e.g., Singh et al. (1986), Key et al. (2000), Taylor and Adelman (2003), and Barrett (2008)).*"
>
> In all these markets, there are no *two* distinct populations of *buyers* and *sellers*.
> There is simply *one* population of *traders*.
>
> This naturally leads to the assumption that the traders' valuations come from a distribution modeling fluctuation of the market value of the item on sale.
>
> To elaborate further, the reader might wonder: "*What if there are multiple segments of the population, each with its own skewed idea of the market price of the item*".
> If at every round, two (possibly different) population segments are drawn i.i.d. and then the valuations and the two traders are drawn i.i.d. from the (possibly different) valuations distribution of each segment, would our techniques still apply?
> The answer is yes.
> To see it, simply note that the setting we investigate is context-free, i.e., we *do not* assume that any information about the traders that come each round is available to the learner before they propose the current trading price at each round.
> Without knowing *a priori* the segment each trader belongs to, writing the expected utility explicitly shows that the problem becomes equivalent to our formulation, where the single distribution is the mixture of all the distributions coming from different segments.
>
> Therefore, the same-distribution assumption in a context-free setting where traders are allowed to buy or sell depending on current market conditions is essentially done without loss of generality.
>
> In a contextual setting, instead, things changes, and different techniques are needed depending on the assumptions relating contextual information to market prices. A (very) recent attempt at investigating a contextual *linear* model for the brokerage problem can be found in [2].
>
> [1] Sherstyuk et al., Randomized double auctions: gains from trade, trader roles, and price discovery.
>
> [2] Bachoc et al., A contextual online learning theory of brokerage.

---

### Meta-Review · Area_Chair_yM2e · 2024-12-23

**Metareview:**

The paper studies a online learning problem for the brokerage who aims to maximize the total trading volume by posting a single price to two sequentially arriving traders. The objective of maximizing total trading volume is new comparing to the classical literature. The algorithmic idea is clean, and the upper bounds and lower bounds derived in this paper are tight. The motivation and intuition is well explained. We thus recommend acceptance.

**Additional Comments On Reviewer Discussion:**

All reviewers in favor of acceptance.

---

### Decision · Program_Chairs · 2025-01-22

Accept (Poster)